# Consistency Regularization for Domain Generalization with Logit Attribution Matching

Han Gao[1,2*]    Kaican Li[2*]    Weiyan Xie[2*]    Zhi Lin[2]    Yongxiang Huang[1]

Luning Wang[1]    Caleb Chen Cao[2]    Nevin L. Zhang[2*]

[1]Huawei Hong Kong AI Framework & Data Technologies Lab
[2]The Hong Kong University of Science and Technology

## Abstract

Domain generalization (DG) is about training models that generalize well under domain shift. Previous research on DG has been conducted mostly in single-source or multi-source settings. In this paper, we consider a third, lesser-known setting where a training domain is endowed with a collection of pairs of examples that share the same semantic information. Such semantic sharing (SS) pairs can be created via data augmentation and then utilized for consistency regularization (CR). We present a theory showing CR is conducive to DG and propose a novel CR method called Logit Attribution Matching (LAM). We conduct experiments on five DG benchmarks and four pretrained models with SS pairs created by both generic and targeted data augmentation methods. LAM outperforms representative single/multi-source DG methods and various CR methods that leverage SS pairs. The code and data of this project are available at https://github.com/Gaohan123/LAM.

## 1 INTRODUCTION

Deep learning models are successful under the independent and identically distributed (i.i.d.) assumption that test data are drawn from the same distribution as training data. However, models that generalize well in-distribution (ID) may be generalizing in unintended ways out-of-distribution (OOD) [Szegedy et al., 2013, Shah et al., 2020, Geirhos et al., 2020, Di Langosco et al., 2022, Yang et al., 2023]. Some image classifiers with great ID performance, in fact, rely on background and style cues to predict the class of foreground objects, leading to poor OOD performance [Beery et al., 2018, Zech et al., 2018, Xiao et al., 2020, Geirhos et al., 2020]. Such reliance on spurious correlations hinders

model performance under domain shift, affecting many real-world applications where the i.i.d. assumption cannot be guaranteed [Michaelis et al., 2019, Alcorn et al., 2019, Koh et al., 2021, Ali et al., 2022, Li et al., 2022].

*Domain generalization* (DG) deals with the conundrum of generalizing under domain shift. Previous research on DG has mostly focused on the single-source and multi-source settings [Zhou et al., 2022, Wang et al., 2022b]. The single-source setting [Volpi et al., 2018, Hendrycks and Dietterich, 2019] is the most general but also the most challenging setting where the domain of a datum is *a priori* unknown. The lack of domain information makes it difficult to tell apart features that are invariant to domain shifts from those that are not. The multi-source setting [Blanchard et al., 2011, Muandet et al., 2013, Ganin et al., 2016, Arjovsky et al., 2019], on the other hand, assumes that such information is available to the degree that every datum is associated with a coarse domain label. Even so, however, it may require a prohibitively large number of diverse domains to solve real-world DG problems [Wang et al., 2024].

In this paper, we study a third lesser-known setting where a training domain is associated with a collection of pairs of examples that share the same semantic information. Such *semantic sharing (SS) pairs* can be created effortlessly using existing data augmentation (DA) methods, as demonstrated by the examples in Figure 1. Given a collection of SS pairs, the task is then to use them to reduce the dependence on spurious correlations.[1] There are several previous DG methods that exploit SS pairs for this purpose [Hendrycks et al., 2020, Mitrovic et al., 2021, Heinze-Deml and Meinshausen, 2021, Mahajan et al., 2021, Robey et al., 2021, Ouyang et al., 2021, Wang et al., 2022c]. They leverage SS pairs via

---

*Equal contribution, listed in alphabetical order.

[1]At a high level of abstraction, this task is related to large language model (LLM) alignment where a collection of preference pairs is used to align an LLM to human intent [Ouyang et al., 2022b, Rafailov et al., 2023]. *In both tasks, the pairs contain information about ideal model behavior that is absent from the training data.* In this sense, one might say that what SS pairs is to domain generalization that preference pairs are to LLM alignment.

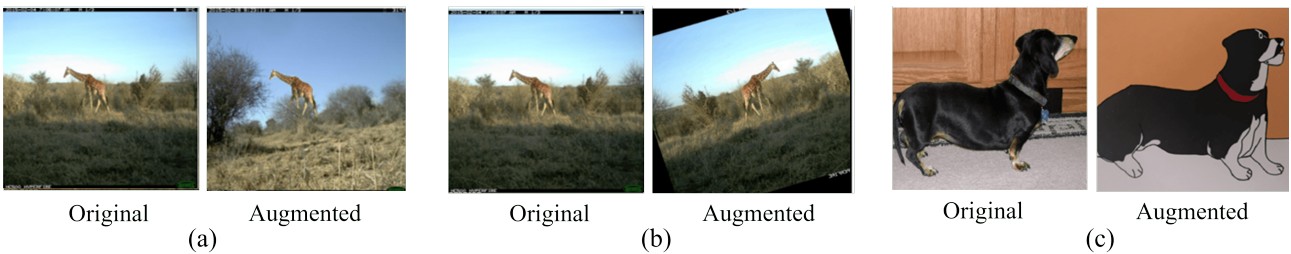

|  Original | Augmented |  | Original | Augmented |  | Original | Augmented |
| :---: | :---: | :---: | :---: | :---: | :---: | :---: | :---: |
| (a) | | | (b) | | | (c) | |

Figure 1: A semantic sharing (SS) pair involves an original training example and a transformed version of it obtained by data augmentation (DA). The examples in the first two pairs share the same semantic information for the "giraffe" class, and the examples in the last pair share the same semantic information for the "dog" class. The augmented example in (a) is created manually via Copy-Paste [Gao et al., 2023], the one in (b) is created using a DA method called RandAugment [Cubuk et al., 2020], and the one in (c) is created using Stable Diffusion [Rombach et al., 2022] (see Appendix C for more details).

*consistency regularization* (CR), a technique proposed in the semi-supervised learning literature to encourage similar predictions on similar inputs [Bachman et al., 2014, Zhang et al., 2020, Chen et al., 2020, Caron et al., 2021]. One drawback they share is that they regard an SS pair $(x, \tilde{x})$ as *unlabeled* and assume $x$ and $\tilde{x}$ contain the same semantic information for *all classes*. As illustrated in Figure 1, however, an SS pair is often created to preserve the semantic information of *one particular class*, and is hence *labeled*. In this paper, we mainly study the use of labeled SS pairs for domain generalization.

We make three contributions in this paper: 1). We present a theory to motivate the use of SS pairs for optimal domain generalization through causally invariant prediction; 2). We propose a novel method called Logit Attribution Matching (LAM) that leverages labeled SS pairs; 3). We empirically demonstrate the advantages of LAM over representative single-source and multi-source DG methods, as well as various CR methods that leverage unlabeled SS pairs.

LAM consistently outperforms previous methods across multiple benchmarks. Take the iWildCam2020-WILDS dataset [Koh et al., 2021] as an example. ERM achieves 30.2% OOD (Macro F1) score on an ImageNet pretrained ResNet-50 model [He et al., 2016]. The score increases to 33.8% when the augmented examples created by RandAugment [Cubuk et al., 2020] are simply added to the training set. It further increases to 36.4% when LAM is applied to the resulting SS pairs. For the augmented examples created by a more sophisticated data augmentation method [Gao et al., 2023], the OOD score is 36.5% when the augmented examples are simply added to the training set. It further increases to 41.2% when LAM is applied to the resulting SS pairs. In this case, the OOD performance increases by $41.2 - 30.2 = 11\%$, with $41.2 - 36.5 = 4.7\%$ due to the exploitation of SS pairs. On CLIP ViT-L/14@336 [Radford et al., 2021], LAM improves the state-of-the-art fine-tuning method from 47.1% to 48.7%. It is hoped that our work can inspire the development of better SS pair creation methods so as to further boost OOD performance of models.

## 2 RELATED WORK

**Domain generalization (DG)** is a fundamental problem in machine learning and has attracted much attention in recent years. A large number of methods have been proposed. In this section, we briefly review several representative methods that are frequently used as baselines in the literature. They are also used in our experiments as baselines.

Most DG methods assume access to multiple training domains [Blanchard et al., 2011, Muandet et al., 2013]. Among those *multi-source* methods, Group Distributionally Robust Optimization (GDRO) [Sagawa et al., 2020] seeks to minimize the worst-case risk across all possible training domains. Invariant Risk Minimization (IRM) [Arjovsky et al., 2019] regularizes ERM with a penalty that enforces cross-domain optimality on the classifier. Variance Risk Extrapolation (V-REx) [Krueger et al., 2020] penalizes the variance of risks in different training domains. Domain-Adversarial Neural Networks (DANN) [Ganin et al., 2016] aims at mapping inputs from each training domain to an invariant distribution in the feature space from which the original domains are indistinguishable.

*Single-source DG* does not assume access to multiple training domains [Volpi et al., 2018, Hendrycks and Dietterich, 2019]. One of the main approaches to single-source DG is to discover predictive features that are more sophisticated than simple cues spuriously correlated with labels. Representation Self-Challenging (RSC) [Huang et al., 2020] and Spectral Decoupling (SD) [Pezeshki et al., 2021] are two prominent methods in this direction. SD suppresses strong dependencies of output on dominant features by regularizing the logits. RSC aims to achieve the same goal in a heuristic manner. Another approach to single-source DG is to simply add augmented examples to the training set [Zhang et al., 2017, Cubuk et al., 2020, Gao et al., 2023]. This approach has been shown to improve OOD performance in many cases, because data augmentation exposes a model to more feature variations during training and thereby enhances its capability in dealing with novel domains.

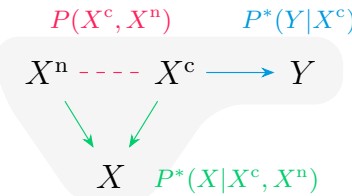

Figure 2: **Causal latent decomposition (CLD) model.** The input of a training example $X$ is generated from two latent variables $X^c$ and $X^n$ which may be statistically correlated due to confounders or direct mechanisms between them. The ground-truth label $Y$ is generated from only $X^c$. The mechanisms that generate $X$ and $Y$ are assumed to be invariant across domains. The corresponding conditional distributions are denoted as $P^*(X|X^c, X^n)$ and $P^*(Y|X^c)$. The joint distribution $P(X^n, X^c)$ of the two latent variables may change across domains. We assume $X^c$ always $d$-separate $Y$ from the other variables.

**Consistency regularization (CR) and semantic sharing (SS) pair creation.** CR encourages a model to make similar predictions on similar inputs. The idea originated from the semi-supervised learning literature [Bachman et al., 2014, Sohn et al., 2020]. It is also used in contrastive learning [Chen et al., 2020] and non-contrastive self-supervised learning [Caron et al., 2021]. In the context of DG, Wang et al. [2022a] conducted a systematic evaluation of various pre-existing CR methods and found that logit matching is most effective with $L^2$-norm (among $L^1$-norm, cosine similarity, etc.). In addition to logit matching with $L^2$-norm, we study a few other options including novel ones such as target-logit matching and LAM which will be discussed in Section 4.1.

To apply CR in the context of DG, we need semantic sharing (SS) pairs. A straightforward way to create SS pairs is to use generic data augmentation (DA) techniques like CutMix [Yun et al., 2019] and RandAugment [Cubuk et al., 2020]. Previous CR methods primarily adopted generic DA techniques [Hendrycks et al., 2020, Xie et al., 2020, Wang et al., 2022a, Chen et al., 2022, Jing et al., 2023, Berezovskiy and Morozov, 2023]. SS pairs can also be created/obtained in ways other than conventional DA. For example, Gao et al. [2023] explored targeted data augmentation (Targeted DA) which utilizes task-specific domain knowledge to augment data. Heinze-Deml and Meinshausen [2021] paired up photos of the same person when analyzing the CelebA dataset [Liu et al., 2015]. For medical images, Ouyang et al. [2022a] created pairs by performing image transformations to simulate different possible acquisition processes. Furthermore, in the case of multiple source domains, SS pairs can be learned. Robey et al. [2021] and Wang et al. [2022c] build image-to-image translation networks between domains and use them to create pairs. Mahajan et al. [2021] propose an iterative algorithm that uses contrastive learning to map images to a latent space, and then match up images from different domains that have the same class label and are close to each other in the latent space.

## 3 A CAUSAL THEORY OF DOMAIN GENERALIZATION

In this section, we present a causal theory of domain generalization, which will be used in the next section to motivate methods for leveraging SS pairs. In the context of DG, a *domain* $d$ is defined by a distribution $P(X, Y)$ over the space of input-label pairs $(X, Y)$. We assume the pairs are generated by the causal model shown in Figure 2.

The model first appeared in Tenenbaum and Freeman [1996], where it is called the *style and content decomposition (SCD) model*, and $X^c$ and $X^n$ are called the *content* and *style* variables respectively. Similar models appeared recently in a number of papers under different terminologies. *The variable $X^c$ denotes the essential information in an image $X$ that a human relies on to assign a label $Y$ to the image.* It is hence said to represent causal factors [Mahajan et al., 2021, Lv et al., 2022, Ye et al., 2022], intended factors [Geirhos et al., 2020], semantic factors [Liu et al., 2021], content factors [Mitrovic et al., 2021], and core factors [Heinze-Deml and Meinshausen, 2021]. In contrast, *the variable $X^n$ denotes the other aspects of $X$ that are not essential to label assignment.* It is hence said to represent non-causal factors, non-intended factors, variation factors, style factors, and non-core factors. As the relationship between $X^c$ and $Y$ does not change across domains, $X^c$ is sometimes said to represent stable features [Zhang et al., 2021], domain-independent factors [Ouyang et al., 2022a], and invariant features [Arjovsky et al., 2019, Ahuja et al., 2021]. In contrast, $X^n$ is said to represent non-stable features, domain-dependent factors, and spurious features.

The term "style" in the SCD model should be understood in a broad sense. In addition to image style, it also includes factors such as background, context, object pose and so on. To avoid confusion, we follow Mahajan et al. [2021], Lv et al. [2022] and refer to $X^c$ and $X^n$ as the *causal and non-causal factors* respectively, and rename the SCD model as the *causal latent decomposition (CLD) model*.

To ground the CLD model, we need to specify three distributions: $P(X^c, X^n)$, $P^*(X|X^c, X^n)$ and $P^*(Y|X^c)$. Together, the three distributions define a joint distribution over the four variables:

$$P(X^c, X^n, X, Y) = P(X^c, X^n)P^*(X|X^c, X^n)P^*(Y|X^c).$$

This joint distribution defines a domain in the CLD framework. We refer to the collection of all such domains for some fixed $P^*(X|X^c, X^n)$ and $P^*(Y|X^c)$ as a *CLD family*.

Let $\mathscr{X}^c$ and $\mathscr{X}^n$ be the sets of all possible values of the

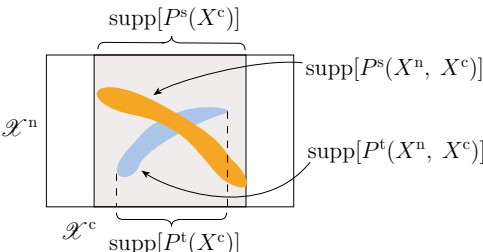

Figure 3: An illustration of conditions for optimal DG under the CLD model. Training examples $x$ are sampled from the latent space, $\mathscr{X}^c \times \mathscr{X}^n$, which we depict as a 2-D box. A prediction model is causally invariant if it makes the same prediction for examples sampled from the same "vertical line" in the latent space. If such a model also minimizes the cross-entropy loss of a source domain, then it makes optimal predictions on all examples $\tilde{x}$ sampled from $\mathrm{supp}[P^s(X^c)] \times \mathscr{X}^n$ (the inner rectangle), not only those from $\mathrm{supp}[P^s(X^c, X^n)]$. This enables optimal generalization to any target domain $P^t$ such that $\mathrm{supp}[P^t(X^c)] \subseteq \mathrm{supp}[P^s(X^c)]$.

latent variables $X^c$ and $X^n$ respectively. Consider an example $x$ generated by $P^*(X|X^c, X^n)$ from a pair of values [2] $(x^c, x^n) \in \mathscr{X}^c \times \mathscr{X}^n$. Let $\tilde{x}$ be another example sampled from the same $x^c$ and a different $\tilde{x}^n$. The two examples $x$ and $\tilde{x}$ contain the same semantic contents and hence should be classified into the same class. In this sense, $x$ and $\tilde{x}$ make up a *semantic sharing (SS) pair*. Let $\hat{P}_\theta(\hat{Y}|X)$ be a *prediction model* with parameters $\theta$. It is said to be *causally invariant* if

$$\hat{P}_\theta(\hat{Y}|X = x) = \hat{P}_\theta(\hat{Y}|X = \tilde{x}), \tag{1}$$

for all SS pairs $(x, \tilde{x})$. In other words, the prediction output does not change in response to variations in the non-causal factors $X^n$ as long as the causal factors $X^c$ remain fixed. Such causal invariance is a key condition for optimal DG.

**Theorem 1 (Conditions for Optimal DG)** *Let $\hat{P}_\theta$ be a prediction model for a CLD family such that different $x^c$ almost always generate different $x$, and let $P^s$ and $P^t$ be a source and a target domain (from the family) such that $\mathrm{supp}[P^t(X^c)] \subseteq \mathrm{supp}[P^s(X^c)]$. Suppose:*

*1). $\hat{P}_\theta$ minimizes the in-distribution (ID) cross-entropy loss $\ell_s(\hat{P}_\theta) = \mathbb{E}_{(x,y) \sim P^s}[-\log \hat{P}_\theta(\hat{Y} = y|x)]$;*

*2). $\hat{P}_\theta$ is causally invariant.*

*Then, the prediction model $\hat{P}_\theta$ also minimizes the out-of-distribution (OOD) cross-entropy loss:*

$$\ell_t(\hat{P}_\theta) = \mathbb{E}_{(x,y) \sim P^t}[-\log \hat{P}_\theta(\hat{Y} = y|x)].$$

*In other words, it generalizes optimally to the target domain.*

---

[2] We use upper case letters to denote variables and lower case letters to denote their values. We use $P$ with variables, e.g., $P(X^c)$, to denote a distribution; and $P$ with variable values, e.g., $P(X^c = x^c)$, to denote a probability value. We may omit the variables if the context is clear, e.g., we may write $P(X^c = x^c)$ as $P(x^c)$.

The proof of this theorem can be found in Appendix A. Closely related theoretical results [Peters et al., 2016, Arjovsky et al., 2019, Mahajan et al., 2021] are discussed in Appendix B. The support $\mathrm{supp}[P(X^c)] = \{x^c \in \mathscr{X}^c \mid P(x^c) > 0\}$ consists of all causal factors that appear in a domain $P$. The assumption on the support between $P^s$ and $P^t$ can be relaxed if we consider approximately optimal DG. We opt for simplicity here since it is not pertinent to the focus of this paper. More importantly, the second condition on $\hat{P}_\theta$ connects consistency regularization (CR) with DG.

The intuition behind Theorem 1 is illustrated in Figure 3. In short, Theorem 1 articulates a set of sufficient conditions for optimal DG. While the causal invariance condition is difficult to verify or fully attain in practice, it can still guide the development of practical DG algorithms. We next discuss CR methods that can bring the model closer to meeting the causal invariance condition.

# 4 CONSISTENCY REGULARIZATION FOR DOMAIN GENERALIZATION

Intuitively, one can make a model more causally invariant by encouraging the model to yield invariant predictions for SS pairs sharing the same $X^c$. So, here is the problem we address in this paper:

*Given a source domain $P^s$ from a CLD family and a set of labeled SS pairs $\{(x_i, \tilde{x}_i; y_i)\}_{i=1}^N$, learn a prediction model $\hat{P}_\theta(Y|X)$ that performs well in any target domain $P^t$ from the same CLD family.*

Recall that a CLD family consists of all the domains defined by the causal model in Figure 2 with fixed $P^*(X|X^c, X^n)$ and $P^*(Y|X^c)$.

## 4.1 CR WITH UNLABELED SS PAIRS

Let us first consider the case where we have a set of *unlabeled* SS pairs $\{(x_i, \tilde{x}_i)\}_{i=1}^N$. The distinction between labeled and unlabeled SS pairs is if the semantic information is invariant for just one particular class or all classes, *not* whether the original examples $x_i$ is labeled. Unlabeled SS pairs contain stronger information than labeled SS pairs: two examples $x_i$ and $\tilde{x}_i$ contain the same semantic information for all classes implies that they contain the same semantic information for every class.

With unlabeled SS pairs, the first two conditions of Theorem 1 can be approximately satisfied by solving the following constrained optimization problem:

$$\min_\theta \quad \mathbb{E}_{(x,y) \sim P^s}[-\log \hat{P}_\theta(\hat{Y} = y|x)]$$

$$\text{subject to} \quad \hat{P}_\theta(\hat{Y}|X = x_i) = \hat{P}_\theta(\hat{Y}|X = \tilde{x}_i), \quad i \in [N].$$

Of course, how well the two conditions are actually satisfied depends on how representative the unlabeled SS pairs we have are of all possible SS pairs.

If we turn the equality constraints into a *consistency regularization (CR)* term, the problem becomes:

$$\min_{\theta}\ \mathbb{E}_{(x,y)\sim P^s}[-\log \hat{P}_\theta(\hat{Y}=y|x)] + \lambda \mathbb{E}_i[r_\theta(x_i,\tilde{x}_i)],$$

where $\lambda$ is a balancing parameter and the summation over $r_\theta(x_i,\tilde{x}_i)$ is a regularization term that relaxes the corresponding equality constraints.

Some notations are needed in order to discuss specific choices for $r_\theta$. Suppose $\hat{P}_\theta$ consists of a feature extractor $f_\phi$ with parameters $\phi$ and a linear classification head $g_w$ with parameters $w$. Hence, $\theta = (\phi, w)$. For an input $x$, let $f_\phi^u(x)$ be the component of the feature vector $f_\phi(x)$ for a feature unit $u$. Let $w_{uy}$ be the weight between a feature unit $u$ and the output unit for a class $y$. The logit for class $y$ is

$$z_\theta^y(x) = \sum_u f_\phi^u(x) w_{uy},$$

where the summation is over all feature units $u$ and the bias is omitted.

For each unlabeled SS pair $(x_i, \tilde{x}_i)$, the CR term $r_\theta(x_i, \tilde{x}_i)$ can be defined in several ways:

$$r_\theta^{\mathsf{KL}}(x_i,\tilde{x}_i) = D_{\mathsf{KL}}\big[\hat{P}_\theta(\hat{Y}|X=x_i) \,\|\, \hat{P}_\theta(\hat{Y}|X=\tilde{x}_i)\big],$$
$$r_\theta^{\mathsf{JS}}(x_i,\tilde{x}_i) = D_{\mathsf{JS}}\big[\hat{P}_\theta(\hat{Y}|X=x_i) \,\|\, \hat{P}_\theta(\hat{Y}|X=\tilde{x}_i)\big],$$
$$r_\theta^{\mathsf{LM}}(x_i,\tilde{x}_i) = \sum_y \big[z_\theta^y(x_i) - z_\theta^y(\tilde{x}_i)\big]^2,$$
$$r_\theta^{\mathsf{FM}}(x_i,\tilde{x}_i) = \sum_u \big[f_\phi^u(x_i) - f_\phi^u(\tilde{x}_i)\big]^2.$$

The first two terms aim to match the output probability distributions of $x_i$ and $\tilde{x}_i$ by minimizing either the KL or JS divergence between them. The third term aims to match their logit vectors, and the fourth term aims to match their feature vectors. They are used in previous methods ReLIC [Mitrovic et al., 2021], AugMix [Hendrycks et al., 2020], CoRE [Heinze-Deml and Meinshausen, 2021], and MatchDG [Mahajan et al., 2021] respectively. Note that while we focus on pairs for simplicity, logit and feature matching can also be extended to the case of multiple examples that share the same semantic contents. To achieve this, we can simply replace the sum of squared differences with the sum of variances. This is done in CoRE and MatchDG.

## 4.2  CR WITH LABELED SS PAIRS

Now consider the case where we have a set of labeled SS pairs $\{(x_i, \tilde{x}_i; y_i)\}_{i=1}^N$. Here, each pair $x_i$ and $\tilde{x}_i$ share the same semantic information only for the class $y_i$. It is no longer justifiable to match all the features, logits or probabilities of all classes. In the following, we propose three methods for leveraging labeled SS pairs.

First, we can match the probabilities or logits of the target class $y_i$ only, leading to what we call *target probability matching (TPM)* and *target logit matching (TLM)*:

$$r_\theta^{\mathsf{TPM}}(x_i,\tilde{x}_i;y_i) = \big[\hat{P}_\theta(\hat{Y}=y_i|x_i) - \hat{P}_\theta(\hat{Y}=y_i|\tilde{x}_i)\big]^2,$$
$$r_\theta^{\mathsf{TLM}}(x_i,\tilde{x}_i;y_i) = \big[z_\theta^{y_i}(x_i) - z_\theta^{y_i}(\tilde{x}_i)\big]^2.$$

To introduce the third method, note that $f_\phi^u(x_i)w_{uy_i}$ is the contribution to the logit $z_\theta^{y_i}(x)$ of $y_i$ from the feature unit $u$. We can match the logit contributions $f_\phi^u(x_i)w_{uy_i}$ and $f_\phi^u(\tilde{x}_i)w_{uy_i}$ from all feature units $u$ to $y_i$. This gives rise to *logit attribution matching (LAM)*:

$$r_\theta^{\mathsf{LAM}}(x_i,\tilde{x}_i;y_i) = \sum_u \big[f_\phi^u(x_i)w_{uy_i} - f_\phi^u(\tilde{x}_i)w_{uy_i}\big]^2.$$

LAM is of finer grain than TLM. Small $r_\theta^{\mathsf{LAM}}$ implies small $r_\theta^{\mathsf{TLM}}$, but not vice versa:

$$r_\theta^{\mathsf{LAM}}(x_i,\tilde{x}_i;y_i) \geq \frac{1}{m}\Big[\sum_u f_\phi^u(x_i)w_{uy_i} - \sum_u f_\phi^u(\tilde{x}_i)w_{uy_i}\Big]^2$$
$$= \frac{1}{m} r_\theta^{\mathsf{TLM}}(x_i,\tilde{x}_i;y_i),$$

where $m$ is the number of feature units. Also, note that

$$r_\theta^{\mathsf{LAM}}(x_i,\tilde{x}_i;y_i) = \sum_u \big[f_\phi^u(x_i) - f_\phi^u(\tilde{x}_i)\big]^2 w_{uy_i}^2.$$

Hence, LAM exerts two complementary regularization forces, one on $g_w$ and the other on $f_\phi$:

1). It encourages the classification head $g_w$ to put large weights $|w_{uy_i}|$ on the feature units $u$ where the values of $x_i$ and $\tilde{x}_i$ are similar, i.e., $f_\phi^u(x_i) \approx f_\phi^u(\tilde{x}_i)$. In other words, *it makes $g_w$ rely on the feature units that reflect the common information contents of $x_i$ and $\tilde{x}_i$.*

2). It encourages the feature extractor $f_\phi$ to make $f_\phi^u(x_i) \approx f_\phi^u(\tilde{x}_i)$ for those feature units $u$ that $g_w$ relies on heavily, i.e., with large weights $|w_{uy_i}|$. In other words, *it encourages $f_\phi$ to channel the common information contents of $x_i$ and $\tilde{x}_i$ toward the units that $g_w$ considers important.*

As $x_i$ and $\tilde{x}_i$ share the causal factors for class $y_i$ but not the non-causal factors, those forces help a model focus more on the causal factors.

## 5  EXPERIMENTS

A direct way to use augmented examples is to add them to the training set and train a model on the combined data using ERM. We denote this approach as ERM+DA. Alternatively, we can pair them up with the original images and apply CR methods on the resulting SS pairs. The main objective of our empirical studies is to compare LAM with ERM+DA,

with ERM itself as a baseline. We also compare LAM with TPM and TLM, as well as previous CR methods.

Another way to utilize the augmented examples is to run a single-source DG algorithm on the combined data. It is also possible to treat the augmented examples as a separate domain and run a multi-source DG algorithm. We further compare LAM with representative single-source and multi-source DG methods in those settings.

Additionally, we assess the impact of the quality and quantity of augmented examples. We consider examples from two DA methods. The first one is RandAugment [Cubuk et al., 2020]. It creates augmented examples by applying a random set of transformations such as resizing, rotating, and color jittering to original images. The second method is Targeted DA [Gao et al., 2023]. It aims to randomize spurious factors while preserving robustly predictive factors. The specific designs of Targeted DA vary across datasets. Targeted DA generally yields more informative SS pairs infused with more specific domain knowledge. We call examples produced from Targeted DA *target-augmented examples*.

## 5.1 DATASETS

Our experiments involve five DG datasets, three with background shifts and two with style shifts.

**iWildCam2020-WILDS (iWildCam)** [Beery et al., 2020, Koh et al., 2021] consists of camera trap photos of animals taken at different locations for wildlife classification. The training domain comprises images from 200 locations, while the test and validation domains contain images from some other locations. Targeted DA is performed by Copy-Paste the animals in a training image to another image (with no animal) taken at a different location where the same animals sometimes appear [Gao et al., 2023].

**ImageNet-9** [Xiao et al., 2020] includes images of nine coarse-grain classes from ImageNet [Deng et al., 2009]. Several synthetic variations are created by segmenting the foreground of each image and place it onto a different background. In our experiments, the synthetic images with a black background are used as target-augmented examples. For the test domain, we use the samples where the foreground of an original image is placed onto the background of a random image.

**NICO** [He et al., 2020] includes around 25,000 images across 19 classes of animals or vehicles in different contexts such as "at home" or "on the beach". As there is no predefined train-test split, we randomly select one context per class for testing and use the remaining contexts for training. Target-augmented training examples and test domains are created in a way similar to ImageNet-9.

**Camelyon17-WILDS (Camelyon)** [Tellez et al., 2018, Koh et al., 2021] contains histopathology images from multiple hospitals for binary tumor classification. Images from three hospitals are used for training, while images from two other hospitals are used for testing and validation respectively. There are stylistic variations among images from different hospitals. One key stylistic difference often observed is the stain color. Therefore, the stain color jitter is applied to training images to create target-augmented examples [Gao et al., 2023].

**PACS** [Li et al., 2017] contains images of objects and creatures in four different styles: *photo*, *art*, *cartoon* and *sketch*. Following common practice [Li et al., 2017, Gulrajani and Lopez-Paz, 2021], we train a model using three of the domains and test the model on the fourth domain. For Targeted DA, we apply Stable Diffusion [Rombach et al., 2022] to images in the *photo* domain to create target-augmented examples in the other three domains. The *photo* domain is therefore not used as the test domain, while the other three domains are used as the test domain in turn. See Appendix C for details.

For all datasets, RandAugment [Cubuk et al., 2020] is performed on all training examples. Targeted DA [Gao et al., 2023] is also performed on all training examples in iWild-Cam and Camelyon. However, it is performed on only about 5% of the training data in ImageNet-9 and NICO, and about 10% of the training data for PACS.

All CR methods have a balancing parameter $\lambda$, which is tuned on the validation domain for iWildCam and Camelyon, and on a test set from the training domain for the other three datasets. For CR and single-source methods, multiple training domains are simply combined into one. More details on how the training data are organized for different types of methods can be found in Table 5 (Appendix E).

## 5.2 NETWORK ARCHITECTURE AND WEIGHT INITIALIZATION

Following Gao et al. [2023], we use a variety of models for different datasets. Specifically, we use an ImageNet pretrained ResNet-50 model [He et al., 2016] for iWildCam, and a randomly initialized DenseNet-121 model [Huang et al., 2017] for Camelyon. We use a CLIP-pretrained ViT-B/16 model [Radford et al., 2021] for ImageNet-9 and NICO, and a CLIP-pretrained ResNet-50 model for PACS.

To showcase the combined use of LAM with advanced CLIP model fine-tuning method can yield SOTA-level performance on iWildCam, we also employ CLIP-pretrained ViT-L/14 and ViT-L/14@336 model for iWildCam.

The use of various model architectures and weight initializations allows us to assess the relative merits of DG algorithms on a mixture of datasets and models. Implementation details about hyperparameters for each dataset and method can also be found in Appendix E.

Table 1: OOD performances of models trained using ERM, ERM+DA, and LAM. The OOD performance of a model is assessed on held-out test domain(s) using Macro F1 score on iWildCam and classification accuracy on all the other datasets. Each model is trained three times, with the standard deviation reported. **Bold** font indicates the best results.

| | | ImageNet-9 (CLIP ViT-B/16) | NICO (CLIP ViT-B/16) | PACS (CLIP ResNet-50) | iWildCam (ImageNet ResNet-50) | Camelyon (DenseNet-121) | Average |
|---|---|---|---|---|---|---|---|
| | ERM | 83.3±1.1 | 95.3±0.1 | 82.8±0.5 | 30.2±0.3 | 65.2±2.6 | 71.4 |
| RandAugment | ERM+DA | 85.3±0.2 | 96.0±0.2 | 83.3±0.3 | 33.8±0.4 | 84.3±2.3 | 76.6 |
| | LAM | 85.6±0.2 | 96.1±0.1 | 83.8±0.4 | 36.4±0.2 | 89.0±1.9 | 78.2 |
| Targeted DA | ERM+DA | 86.0±1.0 | 95.9±0.3 | 84.5±0.5 | 36.5±0.4 | 90.5±0.9 | 78.7 |
| | LAM | **88.1±0.2** | **96.5±0.3** | **86.0±0.3** | **41.2±0.2** | **93.5±1.8** | **81.1** |

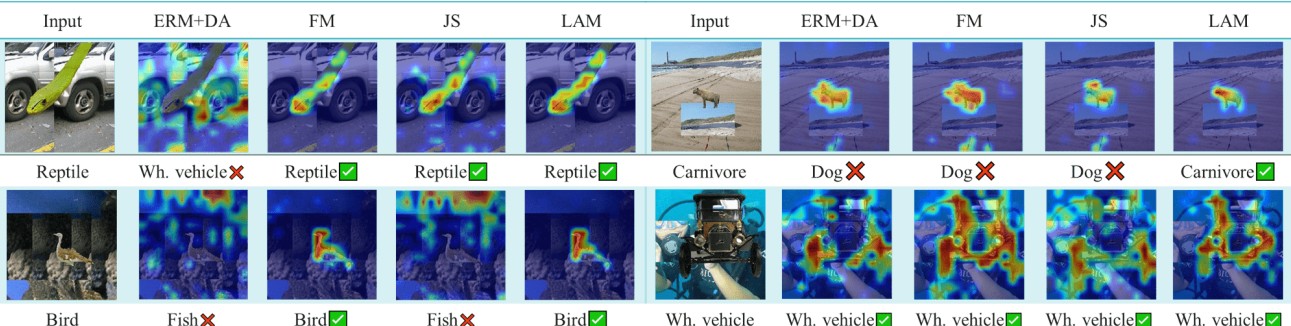

Figure 4: Grad-CAM saliency maps for the top predicted class by models trained on ImageNet-9 using various methods. The model learned using LAM focuses on the foreground objects better.

## 5.3 COMPARISON WITH ERM+DA

Table 1 shows the results for LAM, ERM+DA, and ERM. We see that simply adding augmented data to the training set (ERM+DA) increases the average OOD score from 71.4% to 76.5% with RandAugment [Cubuk et al., 2020], and to 78.7% with Targeted DA [Gao et al., 2023]. *Applying LAM on the resulting SS pairs further increases the scores to 78.2% and 81.1% in the two cases respectively.* In the case of Targeted DA, the average OOD score on those five benchmarks is improved by 81.1-71.4 = 9.7%, with 78.7-71.4 = 7.3% due to data augmentation and 81.1-78.7 = 2.4% due to LAM. The improvements are especially pronounced on the iWildCam and Camelyon datasets, where Targeted DA increases the OOD scores drastically. This is consistent with what was reported in Gao et al. [2023]. LAM further improves the scores by 4.7% and 3.0% respectively.

While trying to gain some insights, we find that LAM makes a model focus on much fewer feature units (see Figure 11 in Appendix F) as compared with ERM+DA. We also use an XAI method called Grad-CAM [Selvaraju et al., 2017] to explain the outputs of the model trained on ImageNet-9 by LAM and they some other methods. Examples are shown in Figure 4 (and Figure 12 in Appendix F). We see that, in all those examples, the LAM model focuses on the foreground objects and gives the correct predictions. Those corroborate

with the analysis we make at the end of Section 4.2. In contrast, the ERM+DA model is more inclined to focus on the wrong part of an input image and predict incorrectly.

In addition to comparing LAM over the traditional ERM which is based on the standard cross-entropy loss, it has been shown in Goyal et al. [2023] that when fine-tuning CLIP models, the use of CLIP contrastive loss with utilizing the CLIP text encoder is more effective. The proposed method is colloquially known as "finetune like you pretrain" (FLYP). In Table 3, we show that the use of LAM can also yield improved OOD performance over FLYP+DA.

## 5.4 IMPACT OF QUALITY AND QUANTITY OF AUGMENTED EXAMPLES

Both LAM and ERM+DA achieve better results with Targeted DA than with RandAugment. We believe this is because Targeted DA generally yields higher quality augmentations than the latter. To further support the claim, we perform additional experiments with ImageNet-9 in the Targeted DA setting. Specifically, we test three different ways to create augmented examples: 1). use a segmentation method called GrabCut [Rother et al., 2004], 2). use another less effective segmentation method called FCN [Long et al., 2015], and 3). simply use bounding boxes that come with ImageNet-9

Table 2: Results for CR methods. **Bold** font indicates best results and arrows indicate changes relative to ERM+DA.

| | | ImageNet-9 | NICO | PACS | iWildCam | Camelyon | Average | iWildCam-N |
|---|---|---|---|---|---|---|---|---|
| RandAugment | ERM+DA | 85.3±0.2 – | 96.0±0.2 – | 83.3±0.3 – | 33.8±0.4 – | 84.3±2.3 – | 76.6 – | 27.6±0.5 – |
| | KL | 85.2±0.3 ↓ | 96.0±0.2 – | 83.1±0.4 ↓ | 34.8±0.2 ↑ | 86.7±5.5 ↑ | 77.2 ↑ | 27.3±0.2 ↓ |
| | JS | 85.2±0.1 ↓ | 95.7±0.5 ↓ | 82.7±1.4 ↓ | 34.5±0.3 ↑ | 83.4±6.7 ↓ | 76.3 ↓ | 26.6±0.4 ↓ |
| | LM | 84.9±0.1 ↓ | 95.8±0.4 ↓ | 82.7±0.2 ↓ | 29.6±0.3 ↓ | 87.9±1.4 ↑ | 76.2 ↓ | 26.5±0.4 ↓ |
| | FM | 85.2±0.1 ↓ | 96.2±0.1 ↑ | 82.3±1.0 ↓ | 31.8±0.3 ↓ | 81.7±5.3 ↓ | 75.4 ↓ | 26.2±0.3 ↓ |
| | TPM | 85.4±0.1 ↑ | **96.2±0.5** ↑ | 82.5±0.6 ↓ | 34.3±0.2 ↑ | 86.9±4.3 ↑ | 77.1 ↑ | 28.0±0.2 ↑ |
| | TLM | 85.3±0.1 – | 95.1±0.3 ↓ | 82.4±0.5 ↓ | 34.1±0.4 ↑ | 87.2±3.2 ↑ | 76.8 ↑ | 27.9±0.4 ↑ |
| | LAM | **85.6±0.2** ↑ | 96.1±0.1 ↑ | **83.8±0.4** ↑ | **36.4±0.2** ↑ | **89.0±1.9** ↑ | **78.2** ↑ | **28.4±0.2** ↑ |
| Targeted DA | ERM+DA | 86.0±1.0 – | 95.9±0.3 – | 84.5±0.5 – | 36.5±0.4 – | 90.5±0.9 – | 78.7 – | 28.2±0.5 – |
| | KL | 86.9±0.2 ↑ | 95.4±0.2 ↓ | 85.0±1.0 ↑ | 40.3±0.3 ↑ | 92.8±1.5 ↑ | 80.1 ↑ | 26.3±0.7 ↓ |
| | JS | 86.0±0.4 – | 95.0±0.3 ↓ | 84.3±0.3 ↓ | 37.1±0.4 ↑ | **94.8±1.2** ↑ | 79.4 ↑ | 25.5±0.6 ↓ |
| | LM | 86.8±0.6 ↑ | 95.3±0.2 ↓ | 83.1±0.8 ↓ | 34.3±0.5 ↓ | 93.4±0.3 ↑ | 78.6 ↓ | 23.9±0.5 ↓ |
| | FM | 87.6±0.1 ↑ | 95.5±0.2 ↓ | 81.7±0.2 ↓ | 36.0±0.3 ↓ | 94.3±0.6 ↑ | 79.0 ↑ | 25.3±0.7 ↓ |
| | TPM | 86.7±0.1 ↑ | 95.8±0.2 ↓ | 84.8±0.7 ↑ | 38.4±0.2 ↑ | 91.7±1.9 ↑ | 79.5 ↑ | 28.3±0.4 ↑ |
| | TLM | 86.2±0.2 ↑ | 95.9±0.2 – | 85.3±1.5 ↑ | 38.5±0.3 ↑ | 93.9±0.7 ↑ | 80.0 ↑ | 28.8±0.2 ↑ |
| | LAM | **88.1±0.2** ↑ | **96.5±0.3** ↑ | **86.0±0.3** ↑ | **41.2±0.2** ↑ | 93.5±1.8 ↑ | **81.1** ↑ | **29.8±0.3** ↑ |

(Box). The resulting OOD scores are as follows:

| Box | | FCN | | GrabCut | |
|---|---|---|---|---|---|
| ERM+DA | LAM | ERM+DA | LAM | ERM+DA | LAM |
| 85.2 | 85.9 | 83.9 | 86.6 | 86.0 | 88.1 |

We see that, as expected, the results with GrabCut are the best, followed by those with FCN and Box, in that order.

We also perform additional experiments with ImageNet-9 to investigate how the quantity of augmented examples influences LAM. Specifically, GrabCut is applied to different percentages of the training examples and LAM is run on the resulting SS pairs. To make a comparison, we do the same thing for the ERM+DA. The resulting OOD scores are as follows:

| | 5% | 10% | 20% | 50% | 100% |
|---|---|---|---|---|---|
| ERM+DA | 86.0 | 86.9 | 86.1 | 87.4 | 87.8 |
| LAM | 88.1 | 88.5 | 88.6 | 89.7 | 90.4 |

It is clear that the increase in the quantity of SS pairs benefits LAM, and the availability of SS pairs for a small fraction of training examples can significantly improve OOD performance already. While providing more SS pairs can also improve the performance of ERM+DA, it is obvious that the improvement is smaller than that of LAM.

## 5.5 COMPARISON WITH OTHER CR METHODS

Table 2 shows the results for LAM and other CR methods. Let us first compare LAM and two other CR methods we

Table 3: Result of finetuning CLIP models with FLYP and LAM on iWildCam. Targeted DA is used here.

| Model | Method | ID F1 | OOD F1 |
|---|---|---|---|
| CLIP-ViT-L/14 | FLYP | 56.9 | 43.4 |
| | FLYP+DA | **59.0** | 44.3 |
| | FLYP+DA+LAM | **59.0** | **45.6** |
| CLIP-ViT-L/14@336 | FLYP | 59.9 | 46.0 |
| | FLYP+DA | 58.9 | 47.1 |
| | FLYP+DA+LAM | **60.9** | **48.7** |

propose in this paper, namely target probability matching (TPM) and target logit matching (TLM). We see that LAM achieves higher OOD scores than the other two methods on average, and it outperforms ERM+DA in all cases while the other two methods do not. Those show that *when making use of labeled SS pairs, it is more effective to apply consistency regularization to the logit contributions of the target classes (LAM) rather than the logits themselves (TLM) or the probabilities of the target classes (TPM).*

Next, we compare LAM with previous strong CR methods, namely probability matching with KL or JS, logit matching (LM) and feature matching (FM). LAM achieves higher OOD scores than those methods on average. Moreover, it achieves the highest score in all cases except for Camelyon with Targeted DA. Moreover, it outperforms ERM+DA in all cases, while the other methods do not. Those show that *it is generally beneficial to regard SS pairs created using both Targeted DA and RandAugment as labeled and apply LAM on them, rather than considering them unlabeled and applying any of the previous CR methods on them.*

Table 4: Results for LAM and representative single-source and multi-source DG methods. **Bold** font indicates the best results and arrows indicate changes relative to ERM+DA.

|  |  | ImageNet-9 | NICO | PACS | iWildCam | Camelyon | Average |
|---|---|---|---|---|---|---|---|
|  | ERM | 83.3±1.1 ↓ | 95.3±0.1 ↓ | 82.8±0.5 ↓ | 30.2±0.3 ↓ | 65.2±2.6 ↓ | 71.4 ↓ |
|  | ERM+DA | 86.0±1.0 – | 95.9±0.3 – | 84.5±0.5 – | 36.5±0.4 – | 90.5±0.9 – | 78.7 – |
| Single-source | RSC | 86.4±0.2 ↑ | 94.0±1.8 ↓ | 84.3±0.6 ↓ | 32.7±0.9 ↓ | 91.6±0.3 ↑ | 77.8 ↓ |
|  | SD | 86.7±0.3 ↑ | 96.0±0.2 ↑ | 85.0±0.4 ↑ | 32.7±0.8 ↓ | **93.5±0.5** ↑ | 78.8 ↑ |
| Multi-source | DANN | 86.5±0.7 ↑ | 95.4±0.7 ↓ | 77.9±1.1 ↓ | 26.0±2.9 ↓ | 90.1±0.9 ↓ | 75.2 ↓ |
|  | GDRO | 83.7±0.8 ↓ | 91.8±1.2 ↓ | 83.5±0.5 ↓ | 37.0±1.0 ↑ | 92.2±0.9 ↑ | 77.6 ↓ |
|  | IRM | 87.1±0.2 ↑ | 93.5±0.2 ↓ | 83.2±0.4 ↓ | 31.7±0.1 ↓ | 90.8±2.6 ↑ | 77.3 ↓ |
|  | V-REx | 83.6±1.4 ↓ | 94.0±0.9 ↓ | 84.4±0.2 ↓ | 35.6±1.6 ↓ | 90.4±4.1 ↓ | 77.6 ↓ |
|  | LAM | **88.1±0.2** ↑ | **96.5±0.3** ↑ | **86.0±0.3** ↑ | **41.2±0.2** ↑ | **93.5±1.8** ↑ | **81.1** ↑ |

In LAM, a labeled SS pair $(x_i, \tilde{x}_i; y_i)$ is used only to regularize the contributions from feature units to the logit of the ground-truth class $y_i$. It does not impact the other classes. In the previous CR methods, on the other hand, the pair is used to regularize the entire feature, logit, or probability vector for $x_i$. It affects other classes as well as $y_i$. This is problematic when a training example $x_i$ contains multiple objects of interest. Some objects that appear in the background of the main object in $x_i$ might be removed during data augmentation. In such a case, the features of those minor objects would be suppressed. To further demonstrate the adverse consequences, we created a variant of the iWildCam dataset [Beery et al., 2020, Koh et al., 2021] by adding a small segmented image of another animal to the background of each image. The new dataset is named **iWildCam-N** (examples of this dataset are given in Appendix D). On this dataset, LAM still improves over ERM+DA. However, the performances of all four previous methods are substantially worse than that of ERM+DA.

Camelyon is a binary classification problem. There is no issue of suppressing features of other classes. This is probably why probability matching with JS is superior to LAM on Camelyon in the case of Targeted DA.

### 5.6 COMPARISON WITH OTHER DG METHODS

Table 4 shows the OOD performances of LAM with six representative single-source and multi-source DG methods reviewed in Section 2. Here only Targeted DA [Gao et al., 2023] is considered. On the first four datasets, LAM outperforms all the six DG methods on average. In particular, it outperforms them by large margins on iWildCam. While LAM improves over ERM+DA on all the first four datasets, the other methods are inferior to ERM+DA in the majority of the cases. On the binary classification dataset Camelyon, however, LAM is on par with SD [Pezeshki et al., 2021], but it still outperforms ERM+DA.

Recall that augmented examples are simply added to the training set for the single-source methods (RSC [Huang et al., 2020] and SD), and they are treated as an additional training domain for the multi-sources methods (DANN [Ganin et al., 2016], GDRO [Sagawa et al., 2020], IRM [Arjovsky et al., 2019] and V-REx [Krueger et al., 2020]). In contrast, LAM applies consistency regularization on the resulting SS pairs. The results in Table 4 show that *consistency regularization with LAM is a more effective way to use augmented examples than representative previous single-source and multi-source DG methods.*

## 6 CONCLUSION

In this paper, we study the setting where a training domain is associated with a collection of example pairs that share the same semantic information. We present a theory to motivate using such semantic sharing (SS) pairs to boost model robustness under domain shift. We find that applying consistency regularization (CR) on the SS pairs, particularly using LAM, significantly improves OOD performance compared to simply adding the augmented examples to the training set. An interesting future direction is to develop more efficient methods for creating more informative SS pairs, e.g., by leveraging advances in generative models. We hope our work could encourage more efforts in manually creating SS pairs for domain generalization, similar to the collection of human preference pairs for LLM alignment.

### ACKNOWLEDGEMENT

We thank the deep learning computing framework MindSpore (https://www.mindspore.cn) and its team for the support on this work. Research on this paper was supported in part by Hong Kong Research Grants Council under grant 16204920. Kaican Li and Weiyan Xie were supported in part by the Huawei PhD Fellowship Scheme.

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

# A PROOFS

**Proof of Theorem 1**: Let us start with the ID cross-entropy loss:

$$\ell_s(\hat{P}_\theta) = \mathbb{E}_{(x,y)\sim P^s}[-\log \hat{P}_\theta(\hat{Y} = y|x)]$$
$$= -\mathbb{E}_{(x^c,x^n)\sim P^s}\mathbb{E}_{x\sim P^*(x|x^c,x^n)}\mathbb{E}_{y\sim P^*(y|x^c)}[\log \hat{P}_\theta(\hat{Y} = y|x)].$$

Because $\hat{P}_\theta$ is causally invariant, $\hat{P}_\theta(\hat{Y} = y|x)$ depends only on $x^c$, but not $x^n$. Denote it as $Q_\theta(\hat{Y} = y|x^c)$. Then, we get

$$\ell_s(\hat{P}_\theta) = -\mathbb{E}_{x^c\sim P^s}\mathbb{E}_{y\sim P^*(y|x^c)}\mathbb{E}_{x^n\sim P^s(x^n|x^c)}\mathbb{E}_{x\sim P^*(x|x^c,x^n)}[\log Q_\theta(\hat{Y} = y|x^c)]$$
$$= -\mathbb{E}_{x^c\sim P^s}\mathbb{E}_{y\sim P^*(y|x^c)}[\log Q_\theta(\hat{Y} = y|x^c)].$$

As the ID loss $\ell_s(\hat{P}_\theta)$ is minimized, the inner expectation is maximized for any $x^c$ such that $P^s(x^c) > 0$.

Now, consider the OOD cross-entropy loss $\ell_t(\hat{P}_\theta)$ of the target domain $P^t$. By symmetry, we have:

$$\ell_t(\hat{P}_\theta) = -\mathbb{E}_{x^c\sim P^t}\mathbb{E}_{y\sim P^*(y|x^c)}[\log Q_\theta(\hat{Y} = y|x^c)].$$

We know from above that the inner expectation is maximized for all $x^c$ such that $P^s(x^c) > 0$. It is also maximized for any $x^c$ such that $P^t(x^c) > 0$ because $\text{supp}[P^t(X^c)] \subseteq \text{supp}[P^s(X^c)]$. $\qquad\square$

# B RELATED THEORETICAL RESULTS

The concept of *causally invariant prediction (CIP)* that we introduce in Section 3 is closely related to a notion described in Peters et al. [2016] that bears a very similar name — *invariant causal prediction (ICP)*. There is a subtle difference. causally invariant prediction refers to the situation where a model makes predictions based on causal factors and, consequently, its performance remains invariant across domains. On the other hand, invariant causal prediction refers to the situation where a model's performance remains invariant across domains and, consequently, its input variables can be considered as causes for the output variable. CIP is for domain generalization while ICP is for causal discovery. In addition, our work involves latent variables ($X^c$ and $X^n$) while Peters et al. [2016] deal with only observed variables.

Our Theorem 1 is closely related to Theorem 1 of Mahajan et al. [2021] and Theorem 3.2 of Arjovsky [2020]. However, the causal model used by Mahajan et al. [2021] has three more latent variables than the one we use. In fact, our model can be viewed as their model with the additional latent variables "integrated out". As such, our theorem targets a more general setting. In addition, their theorem focuses exclusively on feature matching and hence cannot be used to motivate logit attribution matching (LAM). Arjovsky's theorem also focuses on the feature extractor. It requires examples with the same feature representation to have approximately the same output probability distributions under the generative model. In this sense, it seeks to obtain features with invariant prediction by the *generative model*. In contrast, our theorem requires a *prediction model* to be invariant to the non-causal factors. While Arjovsky's theorem is used to motivate a DG algorithm called invariant risk minimization (IRM), our theorem is used to justify consistency regularization.

In this paper, we use a causal theory of domain generalization to motivate consistency regularization methods. It should be noted that there are other theories for domain generalization that are based on divergence between domains [Ben-David et al., 2010, Liu et al., 2020]. Those theories are used to motivate the domain invariant representation approach to domain generalization. However, they cannot be used to justify consistency regularization methods.

# C MORE DETAILS OF SS PAIR CREATION USING TARGETED DA

An SS pair is formed by a training example and an augmented example. The SS pair creation using Targeted DA for each dataset has been introduced in Section 5.1. We provide more details and examples here.

## C.1 IWILDCAM AND IWILDCAM-N

For iWildCam and iWildCam-N, we utilized a Targeted DA technique named Copy-Paste (same-y) from Gao et al. [2023]. This DA method pastes the animal foreground onto a background image sampled from the same habitat where the same

animal species has been observed. There is a category of images labeled "empty" in the iWildCam dataset. These images do not contain any animals and were used as background images when creating augmented examples. We used the segmentation for the animal foregrounds provided by Beery et al. [2021] to apply this DA. Augmented examples produced by this DA approach are provided in Figure 5.

| True label | Training example | Augmented example | True label | Training example | Augmented example |
|---|---|---|---|---|---|
| Giraffa Camelopa-rdalis | | | Loxodonta Africana | | |
| Aepyceros Melampus | | | Crax Rubra | | |

Figure 5: SS pairs created via Copy-Paste (same-y) DA for iWildCam. This DA method involves pasting the animal onto another image without animals sampled from the location where the same animal species has been observed.

## C.2 IMAGENET-9

In our main experiments, the synthetic images with a black background were used as augmented data for ImageNet-9. Those augmented examples were created based on the GrabCut segmentation. As described in Section 5.4, to assess the performance of LAM under augmented examples in various qualities, we also considered the augmented examples created based on the bounding boxes and semantic segmentation. Specifically, we used the bounding boxes provided by the ImageNet [Deng et al., 2009] and semantic segmentation produced via FCN [Long et al., 2015], a semantic segmentation method. Augmented examples in various qualities are given in Figure 6.

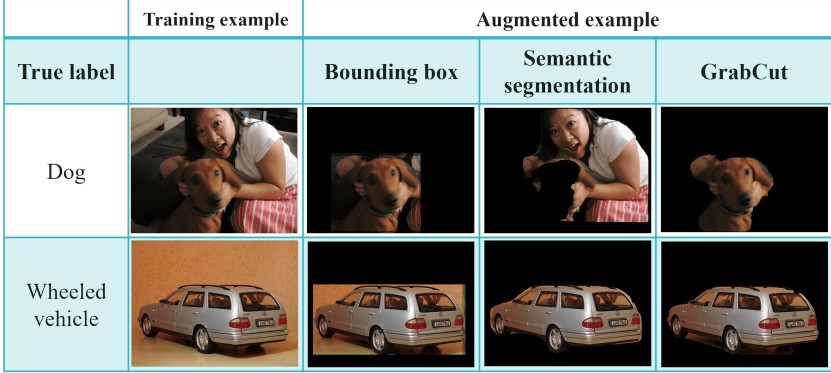

| True label | Training example | Augmented example | | |
|---|---|---|---|---|
| | | Bounding box | Semantic segmentation | GrabCut |
| Dog | | | | |
| Wheeled vehicle | | | | |

Figure 6: Augmented examples in various qualities created for ImageNet-9.

## C.3 NICO

For creating the augmented examples for NICO, we placed the foreground segmentation onto the background of a random image. We used GrabCut [Rother et al., 2004] to identify the foreground segmentation for 20 images in each class of NICO, which constituted about 5% of its training data. On average, the segmentation of an image took us around three seconds.

Since NICO does not have "empty" background images like iWildCam, we had to create synthetic background images. To do this, we removed the foreground in the image by coloring the image region corresponding to the foreground segmentation in black. We created the synthetic background images for all images with the foreground segmentation. When creating

the augmented example, the foreground segmentation in the training example is pasted onto a randomly selected synthetic background image. See Figure 7 for some NICO augmented examples.

Figure 7: SS pairs created for NICO by placing the foreground segmentation onto a randomly selected synthetic background image.

Figure 8: SS pairs created by stain color jitter for Camelyon dataset. This DA randomizes the average stain level in the image.

### C.4 CAMELYON

In dealing with the Camelyon dataset, we adopted the strategy outlined in Gao et al. [2023] to use the stain color jitter [Tellez et al., 2018] as the Targeted DA to create the augmented examples. This technique transforms images by jittering their color in the hematoxylin and eosin staining color space. This DA addresses the style shift associated with the stain color resulting from diverse staining techniques used across different hospitals. It randomizes the average stain level in each image while maintaining all other information as predictive features. Some augmented examples are shown in Figure 8.

### C.5 PACS

To create SS pairs for PACS, we used StableDiffusion v2 [Rombach et al., 2022] to translate images from the *photo* domain of PACS into a different style. Given a training example $x$ of label $y$, we added a mild level of Gaussian noise to the latent representation of $x$, and then removed the noise under the guidance of a text prompt. The prompt we used is "a minimalist drawing of a `class_name`, outline only, no texture" where `class_name` is the name of $y$. We chose this prompt because it produces the best visual quality among what we have explored. Finally, we decoded the generated noise-free latent representation, producing the corresponding augmented example $\tilde{x}$. See Figure 9 for some examples.

| True label | Training example | Augmented example | True label | Training example | Augmented example |
|---|---|---|---|---|---|
| Dog | | | Horse | | |
| House | | | Giraffe | | |

Figure 9: SS pairs created via StableDiffusion that generates augmented example from the training examples of the *photo* domain in the PACS dataset. The prompt we use is "a minimalist drawing of a class_name, outline only, no texture" where class_name is the name of the true class label.

## D    DETAILS OF IWILDCAM-N DATASET

**iWildCam-N** dataset is an altered version of the iWildCam dataset [Beery et al., 2020, Koh et al., 2021], which includes extra background noise in addition to the original background shift in the iWildCam. This additional noise was created by inserting an animal foreground of a different animal species, sampled from a randomly selected image, onto the background of the image. To ensure the main semantic context of the image is not distorted due to the introduced noise, we limited the size of the introduced animal to be smaller than the pre-existing animal foreground and took steps to prevent overlap between the newly incorporated animal and the original animal foreground. We applied this operation on all images in the iWildCam dataset except for the images in the "empty" category, which do not contain any animals. The "empty" category was also excluded from the iWildCam-N dataset.

In Figure 10. We provide some examples of the iWildCam-N and their original images in the iWildCam to illustrate the background noise introduced in iWildCam-N.

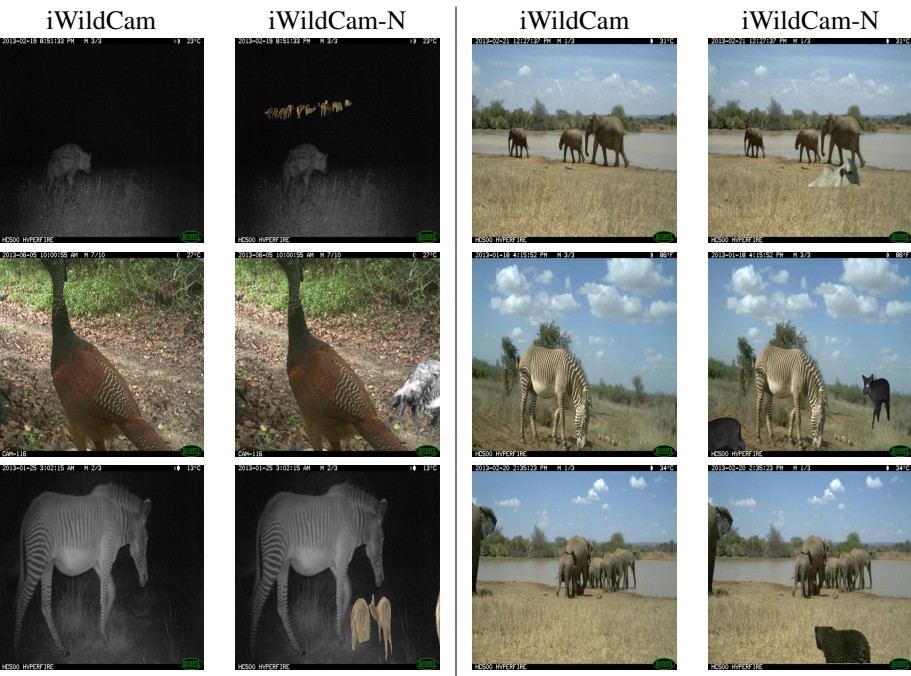

Figure 10: Sample images in iWildCam-N. The background noise is created by adding other small animals to the background of each image.

# E   ADDITIONAL IMPLEMENTATION DETAILS

The use of augmented examples in different methods, including in the ERM+DA, CR-based DG methods, and other multi-source and single-source methods, has been introduced in Section 5. We provide a summary in Table 5.

Table 5: The use of training data in different methods.

| Category | Methods | Training data | Remark |
|---|---|---|---|
| Baseline | ERM | training examples | - |
| ERM+DA & Single-source | ERM+DA RSC, SD | training examples + aug. examples | As additional training data, augmented examples are combined with training examples to train the model. |
| CR-based | LAM, KL, JS, LM, FM TLM, TPM | training examples + aug. examples | The training examples are paired with augmented examples to train the model. |
| Multi-source | DANN, GDRO IRM, VREx | $d_1$: training examples $d_2$: aug. examples | Training examples are regarded as one domain; augmented examples form another domain. |

All experiments were conducted on a single NVIDIA V100 GPU. For ImageNet-9, NICO, and PACS, we used the two-step training strategy of linear probing and then full finetuning (LP-FT) [Kumar et al., 2022], while for other datasets we did normal finetuning. The summary of the hyperparameter setting is shown in Table 6.

Table 6: Hyperparameter setting for all the main experiments. SS pair transformation refers to the transformation applied to training examples and corresponding augmented examples while training. For other DG methods, we use the default hyperparameters provided by DomainBed [Gulrajani and Lopez-Paz, 2021] as the initial values, followed by a hyperparameter tuning process. "bs" stands for batch size.

| Dataset | ImageNet-9 & NICO | | PACS | iWildCam | Camelyon |
|---|---|---|---|---|---|
| Model | CLIP ViT-B/16 | | CLIP ResNet-50 | ResNet-50 | DenseNet-121 |
| Pretrained | ImageNet pretrained | | | | False |
| Image Size | [224, 224] | | | [448, 448] | [96, 96] |
| LAM/ Logit Match (LM)/ Prob. Match (KL) | LP/FT epochs: 10/20 | | LP/FT epochs: 10/40 | epochs: 20 | epochs: 10 |
| | LP/FT learning rate: 0.003/3e-5 | | | learning rate: 3.49e-5 | learning rate: 3.07e-3 |
| | LP/FT training bs: 128/64 LP/FT SS pair bs: 256/64 | | LP/FT training bs: 48/48 LP/FT SS pair bs: 32/32 | training bs: 10 SS pair bs: 10 | training bs: 128 SS pair bs: 128 |
| | $\lambda = 10$ | $\lambda = 0.5$ | $\lambda = 0.2$ | $\lambda = 5$ (LAM, KL) $\lambda = 0.05$ (LM) | $\lambda = 10$ (LAM) $\lambda = 1$ (LM, KL) |
| | SS pair transform: RandCrop RandHorizontalFlip Normalize | | SS pair transform: RandCrop RandHorizontalFlip ColorJitter RandGrayscale Normalize | SS pair transform: Normalize | SS pair transform: Normalize |
| | N/A | | $p = 0.9$ | N/A | |
| Feature Matching (FM) | $\lambda = 0.01$ | | | $\lambda = 0.05$ | $\lambda = 0.1$ |
| Prob. Match (JS) | FT training bs: 32 FT SS pair bs: 48 | | FT training bs: 48 FT SS pair bs: 48 | FT training bs:10 FT SS pair bs: 20 | FT training bs: 128 FT SS pair bs: 128 |
| Other Methods | LP/FT training bs: 128/64 | | LP/FT training bs: 48/48 | training bs: 24 | training bs: 128 |

# F VISUALIZATIONS ABOUT THE EFFECTS OF LAM

In Section 4.2, we have argued that LAM exerts two complementary regularization forces, one on the feature extractor and another on the classification head. In combination, they encourage a model to focus on the causal factors when making predictions.

To provide some empirical evidence for the claim, we show in Figure 11 the weight distributions of the classification heads of three models trained on the ImageNet-9 dataset. We see that the LAM model has significantly fewer high weights than those of the other two models. This indicates that the LAM is indeed more "focused" than the other models.

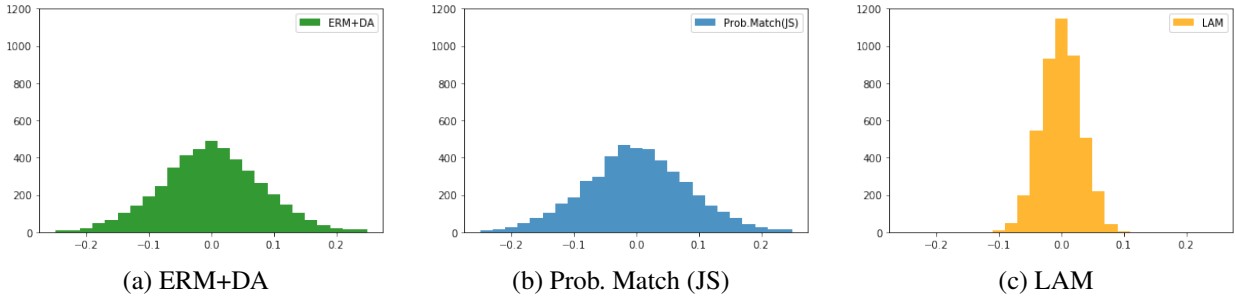

(a) ERM+DA  (b) Prob. Match (JS)  (c) LAM

Figure 11: Distributions of the weights of the classification heads of the models learned using ERM+DA, Probability Matching (JS), and LAM on ImageNet-9 dataset.

What does the LAM model focus on? Visual examples in Figure 4 indicate that it focuses on the foreground objects. This claim is also supported by the additional examples in Figure 12.

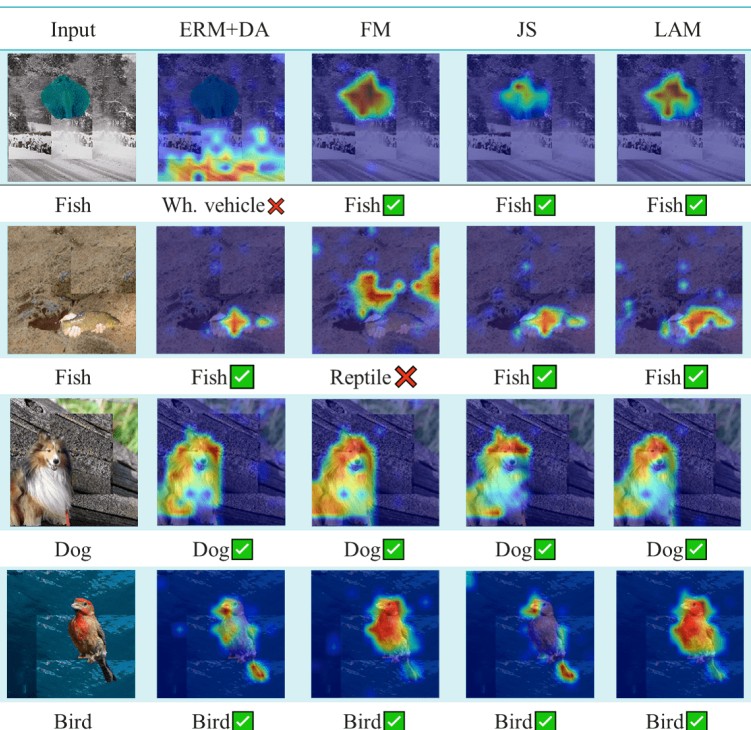

Figure 12: GradCAM saliency maps for the top predicted class by models trained on ImageNet-9 using various methods. The model learned using LAM focuses on the foreground objects better.