# OpenReview forum: "Consistency Regularization for Domain Generalization with Logit Attribution Matching"
_auai.org/UAI/2024/Conference — UAI 2024 poster_

### Official Review · Reviewer_97yb · 2024-03-20

**Q2-1 Originality-Novelty:** 2
**Q2-2 Correctness-Technical Quality:** 3
**Q2-5 Clarity Of Writing:** 3

**Q1 Summary And Contributions:**

The authors propose improving unsupervised domain adaptation (UDA) using semantic sharing (SS) pairs, pairs of inputs that share any factors that are related to the outcome variable but may differ in other factors. They give conditions under which learning using SS pairs leads to optimal adaptation and give an algorithm inspired by this idea that uses constructed SS pairs from data augmentation procedures. The approach is evaluated on several image classification UDA tasks and shows promising performance compared to using ERM with data augmentation and established UDA baselines.

**Q2-3 Extent To Which Claims Are Supported By Evidence:**

3: Good: the main claims are supported by convincing evidence (in the form of adequate experimental evaluation, proofs, (pseudo-)code, references, assumptions).

**Q2-4 Reproducibility:**

3: Good: key resources (e.g. proofs, code, data) are available and key details (e.g. proofs, experimental setup) are sufficiently well-described for competent researchers to confidently reproduce the main results.

**Q3 Main Strengths:**

* The paper is easy to follow and makes its contributions clear.
* The empirical evaluation is well suited for the paper and includes relevant baselines and multiple datasets which fit the structure of the motivating problem.
* The approach is simple and the authors investigate several possible variants of the regularisation penalties.
* The authors acknowledge limitations of their work regarding the scope and strength of assumptions.

**Q4 Main Weakness:**

* Some experimental details are missing from the paper
* The "causal theory of domain generalization" boils down to a single conditional independence assumption which has been made in previous work. I would tone that contribution down.
* Any discussion on the take-aways from analysis and experiments is very light

**Q5 Detailed Comments To The Authors:**

* The assumptions necessary for Theorem 1 questions the need for solving UDA at all when the conditions hold.
  1. A model being "causal-invariant" is unverifiable without also assuming overlapping supports for the non-causal factors, i.e., expanding condition 3) to the entire X.
  2. To arrive at a "causal-invariant" model, we must be able to create SS pairs that span this space. If we can't, we no longer have a guarantee.
  3. If we believe that we can create SS pairs that span all possible variations of the non-causal factors, then we are effectively assuming that we can label the target domain.
  4. But if we can label the target domain, why do we need UDA in the first place?
The authors discuss the plausibility of the assumptions right before equation (3), but I would have liked to see an extended analysis of this issue.


* The point above could partially explain why TargetedDA works much better than RandAugment. And I believe that it is also part of the reason we see ERM+DA being close to LAM in many of the empirical results: If we can sample augmentations from a distribution that covers the target domain, ERM+DA should do just fine. (But the augmentation itself is not a contribution of this work). On this note, how many augmented samples (SS pairs) were generated for each of the data sets? Does the empirical performance depend on these numbers? For example, does ERM+DA improve if more augmented samples are used?


* The "causal theory of domain generalization" completely ignores unobserved confounding and does not make an explicit assumption that no such confounding exists. For example, if there was a variable $U$ which influenced both non-causal factors $X^n$ and the outcome $Y$, the model in Figure 2 would be invalid. This is unlikely specifically in the object detection/classification tasks considered in this work, but could well happen in more general settings (and Section 3 is not presented as pertaining only to object classification). For example, in a medicine, an image learning task could be to predict the mortality of a patient based on a chest X-ray. It could be that the hospital ($U$) determines which X-ray machine is used (impacting the style of $X^n$) and some hospitals offer better care than other hospitals (therefore affecting $Y$). However, it would be strange to say that the X-ray machine used is **causal** of recovery/mortality. Thus, $U$ is a confounder.


* In experiments, labelled SS pairs offer no consistent gain over ERM+DA, especially not when Targeted DA is used. Would the authors recommend against using this methodology?


* In most experiments (see 5.2), the authors use models that have been pre-trained on image learning tasks (classification/self-supervision). While most likely useful in practice, this muddies the separation between source and target domains somewhat, since it becomes harder to control what labeled images have been "observed" by the pre-trained representations. Perhaps the authors can comment on this?


* The first example in Figure 4 points to the difficulty in creating SS pairs by data augmentation without destroying the label information. The authors claim that ERM+DA predicts the wrong class when it outputs "Wheeled vehicle", but there is clearly a wheeled vehicle in the image. It may not be the label the authors intended, but it does not appear wrong to me either. I would have liked to see a discussion of this issue.


* In Table 5, the authors state that "for other DG methods, the method-specific hyperparameters follow the default setting in DomainBed [Gulrajani and Lopez-Paz, 2021]" whereas "All CR methods have a balancing parameter λ, which is tuned on the validation domain for iWildCam and Camelyon, and on a test set from the training domain for the other three datasets." Is this a fair comparison? On a related note, was the same network architecture used for all baselines (E.g., DANN, IRM) as well?

### Minor points

* The term "causal-invariant" is not very transparent given that the model is not invariant to changes in the causal factors---quite the opposite. If anything, the model is invariant to spurious (non-causal) correlations. Invariance to spurious correlations/associations is terminology that has been used before for similar concepts.

**Q9 Complying With Reviewing Instructions:**

Yes

---

> ### Author Rebuttal · Authors · 2024-04-06
>
> We are grateful for the reviewer's time and insightful feedback. Regarding the concerns and questions the reviewer raised, we provide detailed explanations below.
> ## Necessity/utility of UDA
> Yes, in practice the causal-invariant assumption is hard to achieve or verify. We need UDA because we want our models to be as robust as possible even without access to target domains. It is okay if we can’t *perfect* them as long as the models are made more reliable.
>
> In general, the more invariant a model is to a certain set of $x^n$, the more likely it is also more invariant to a wider set of $x^n$, assuming there are some basic regularities in the model, e.g., continuity. If we can make additional assumptions about the target domains of specific tasks, which we usually can (to some degree), then it is more likely for the invariance to generalize even further.
>
> Indeed, there is no *guarantee* for optimal DG if the model did not “observe” all possible $x^n$ during training. Our theorem merely states a set of ideal conditions where optimal DG can be guaranteed. It serves as a theoretical objective to guide the design of practical algorithms.
>
> We thank the reviewer for the insightful comments on the necessity of UDA. We will fit the above discussion into our paper if the reviewer finds it reasonable. Finally, as the reviewer kindly suggested, we will also tone down our theoretical contribution and clearly point out similar results in the literature (e.g., [1]).
>
> [1] Representation Learning via Invariant Causal Mechanisms.
> ## LAM vs ERM+DA
> Ideally, through data augmentation (DA), we hope to eliminate the mutual information $I(X^n, Y)$ so that ERM has no incentive to rely on $X^n$ to predict. In practice, however, DA is often far from perfect, and we don't know how to get $I(X^n, Y)$ sufficiently small. If $I(X^n, Y)$ is not sufficiently small, then it is unlikely for ERM+DA to learn to be causal-invariant by itself, i.e. it would *not* do just fine.
>
> In comparison, LAM can improve the situation as it helps explicitly enforce causal-invariant prediction. This explains why LAM often outperforms ERM+DA despite using the same data.
>
> To “sample augmentations from a distribution that covers the target domain”, one must assume that *near-perfect* augmentations are already available. In reality, however, this is often impossible or prohibitively expensive because it would require: (1) a DA method that can perfectly randomize $x^n$ while keeping $x^c$ unchanged; and (2) access to all possible $\tilde{x}^n$. Targeted DA is probably better than RandAugment on both fronts, but apparently, it is still not perfect.
>
> Although it is hard to perfect DA, we can design CR methods such as LAM to better utilize the imperfect DA. As we have shown in the paper, LAM consistently outperforms ERM+DA in both RandAugment and Targeted DA cases. In fact, Table 1 shows that the advantage of LAM even enlarges with better DA. Of course, if we continue to improve DA, ERM+DA will start to and eventually catch up with LAM, but this is unlikely to happen soon for complex real-world tasks. Meanwhile, LAM offers an easy way to boost OOD performance with just moderate, inexpensive DA.
> ## Question related to amounts of SS pairs used
> This detail of the amounts of SS pairs used for each dataset was provided in the last paragraph of Section 5.1 of our paper. Both ERM+DA and LAM benefit from more SS pairs, while LAM benefits more. We used ImageNet-9 as an example and tried different amounts of SS pairs (from Targeted DA). The resulting OOD accuracies are shown below:
> |#SS pairs (in % training examples)|ERM|ERM+DA|LAM|
> |-|-|-|-|
> |0%|83.3|N/A|N/A|
> |5%|N/A|86.0|88.1|
> |10%|N/A|86.9|88.5|
> |20%|N/A|86.1|88.6|
> |50%|N/A|87.4|89.7|
> |100%|N/A|87.8|**90.4**|
> ## About ignored confounding
> Thanks for the insightful comment! We agree there may be unobserved confounding between $X^n$ and $Y$ sometimes. Fortunately, this scenario can be easily incorporated into our theory by *slightly* modifying the first condition and the result of Theorem 1.
>
> Specifically, we can change the first condition into: $\hat{P}_θ$ has the minimal ID cross entropy loss *among all causal-invariant predictors*. This is because when there is confounding, $X^n$ and $Y$ are no longer d-separated by $X^c$, and therefore the ID loss minimizer may not be causal-invariant, leading to a potential contradiction of the first two original conditions. Changing the first condition as mentioned resolves the conflict. Accordingly, the result would become: $\hat{P}_θ$ also minimizes the OOD cross entropy loss *among all causal-invariant predictors*. This hardly weakens the strength of the result. In addition, the proof of Theorem 1 can be mostly reused since it did not require the d-separation to hold.

---

### Official Review · Reviewer_p8b3 · 2024-03-20

**Q2-1 Originality-Novelty:** 3
**Q2-2 Correctness-Technical Quality:** 4
**Q2-5 Clarity Of Writing:** 4

**Q1 Summary And Contributions:**

This paper provide a unified view of domain generalisation from a causal perspective, and propose a novel method called LAM and this methods has been tested on a wide range of benchmark datasets for domain generalisation tasks. The experimental results proof the effectiveness of LAM method. The main contribution include: (1) the analysis (2) the LAM model and (3) experiments.

**Q2-3 Extent To Which Claims Are Supported By Evidence:**

3: Good: the main claims are supported by convincing evidence (in the form of adequate experimental evaluation, proofs, (pseudo-)code, references, assumptions).

**Q2-4 Reproducibility:**

4: Excellent: key resources (e.g. proofs, code, data) are available and key details (e.g. proof sketches, experimental setup) are comprehensively described for competent researchers to confidently and easily reproduce the main results.

**Q3 Main Strengths:**

The overall quality of this paper is very good and I really enjoy reading the paper. The paper start by pointing out the key ingredient: "how to utilise ss pairs which offers a labelled information of (x_{i}, x_{j}) with (y_{i}=y_{j}), not unlabelled x pair only". This difference is further addressed via analysis the domain generalisation problem from a causal view and connect their analysis with existing literature. The analysis naturally leads to the proposed solution, LAM. And the authors tested their methods on various datasets against state of the art methods.

**Q4 Main Weakness:**

I feel this paper will benefit from better presentation, such as the main motivation in section1, right before the contributions, can be elaborated or emphased further. The section of the general theory and methodology (3 and 4) could be merged together.

In the experiment section, it might be interesting to see if multiple data augmentation techniques are applied to create ss pairs altogether, what would be the effect?

**Q5 Detailed Comments To The Authors:**

For the detail comments, in general the paper is well motivation and very solid for theoretical analysis and experiments. I personally think the presentation could be improved further to help readers understanding this paper better. Some part of the experiments could be added to understand the influence of multiple DA technique to create SS pairs. (such as heterogenous X_{n} -- X_{c}).

**Q9 Complying With Reviewing Instructions:**

Yes

---

> ### Author Rebuttal · Authors · 2024-04-07
>
> We are grateful for the reviewer's time and insightful feedback. It is encouraging to know that the reviewer found our work is with “well motivation” and is “very solid for theoretical analysis and experiments”. We appreciate that the reviewer has thoroughly engaged with our work and acknowledged its merits.
>
> Regarding the comments and suggestions of the reviewer, we provide detailed response below.
> ## Better presentation
> Thank you for your kind suggestion. We will take your feedback into consideration and refine the motivation in Section 1 to provide a clearer and more compelling argument for the relevance and importance of our work. Regarding the organization of Sections 3 and 4, we agree with you that the connection between theory and methodology could be improved. To address this, we will revise the sections to create a more intuitive transition that highlights how our theoretical insights underpin the consistency regularization (CR) approaches. We agree that in the current version, the connection may appear vague, and we intend to clarify this relationship.
> ## Applying multiple data augmentation techniques
> “In the experiment section, it might be interesting to see if multiple data augmentation techniques are applied to create ss pairs altogether, what would be the effect?”
>
> As you suggested, we have tried to use the RandAugment [1] and targeted data augmentation (targeted DA) [2] together to create augmented examples on iWildCam. To be more specific, for a training image $x_i$, we created two augmented examples $\tilde{x}_i^1$ (from RandAugment) and $\tilde{x}_i^2$ (from targeted DA). The LAM regularization is then applied to both the pairs, $(x_i, \tilde{x}_i^1; y_i)$ and $(x_i, \tilde{x}_i^2; y_i)$.
>
> The experiment results are shown below, where the results of the first 6 rows are copied from our paper and results in the last 2 rows are new (ERM: without using DA; ERM+DA: simply adding the augmented examples in the ERM training). From the results, we can see that when using the augmented examples from both DA, the OOD performance of LAM can be further improved, yielding 42.1 now. We believe this improvement is largely due to the heterogeneity of the augmented examples as the reviewer mentioned. Through learning from a more diverse range of augmented examples, the model has become even more robust to the variation of $X^n$.
> | Augmentation|Method|OOD F1 $\uparrow$|
> |-|-|-|
> |No|ERM|30.2$\pm$0.3|
> |RandAugment|ERM+DA|33.8$\pm$0.4|
> |RandAugment|LAM|36.4$\pm$0.2|
> |Targeted DA|ERM+DA|36.5$\pm$0.4|
> |Targeted DA|LAM|41.2$\pm$0.2|
> |RandAugment+Targeted DA| ERM+DA|36.9$\pm$0.3|
> |RandAugment+Targeted DA| LAM|**42.1$\pm$0.4**|
>
> We thank the reviewer for pointing out the potential of using multiple data augmentation techniques to create SS pairs, and we will include the results in our paper.
> ## Additional experiment results
> In addition, current SOTA-level DG methods are mostly model ensemble methods [3,4,5] and fine-tuning techniques for pre-trained models [6,7,8].  LAM can be seamlessly combined with those methods. We conducted some experiments to see if the combinations can help further push the limit of DG (This is not directly related to the comments in the review, but we hope the reviewer may find it interesting).
>
> Specifically, we combined LAM with a SOTA-level fine-tuning technique, FLYP [6], and compared the results with SOTA model ensemble methods.  FLYP is a technique for finetuning vision-language models like CLIP to better handle downstream classification tasks. We conducted experiments on the iWildCam dataset, and the results are shown below. The performance of FLYP+DA was obtained by incorporating target-augmented images in the FLYP fine-tuning process, while FLYP+DA+LAM is our method which applies the LAM on top of FLYP+DA.
> | Method|Backbone|ID F1$\uparrow$|OOD F1$\uparrow$|
> |-|-|-|-|
> |FLYP+DA+LAM|CLIP ViT-L/14@336|60.9|**48.7**|
> |FLYP+DA|CLIP ViT-L/14@336|58.9|47.1|
> |FLYP [6]|CLIP ViT-L/14@336|59.9|46.0|
> |WiSE-FT [5]|CLIP ViT-L/14@336  | 55.8|46.4|
> |ERM|CLIP ViT-L/14@336|52.1|39.9|
> |||||
> |FLYP+DA+LAM|CLIP ViT-L/14|59.0| **45.6**|
> |FLYP+DA|CLIP ViT-L/14|59.0|44.3|
> |FLYP  [6]|CLIP ViT-L/14| 56.9|43.4|
> |Model Soup [4]|CLIP ViT-L/14|57.6|43.3|
> |ERM|CLIP ViT-L/14|55.8|41.4|
>
> The results show that LAM significantly improves the model's OOD performance over both FLYP and FLYP+DA. In addition, FLYP+DA+LAM (without model ensemble) outperforms all the ensemble methods, Model Soup [4] and WiSE-FT [5].
>
> We plan to also add the results in our paper.
>
> [References](https://www.dropbox.com/scl/fi/1fwchfh94zppz54qijp5o/reference3.pdf?rlkey=dy92hi8hve50mxag7ev76nwkz&dl=0)

---

### Official Review · Reviewer_bXye · 2024-03-21

**Q2-1 Originality-Novelty:** 2
**Q2-2 Correctness-Technical Quality:** 2
**Q2-5 Clarity Of Writing:** 2

**Q1 Summary And Contributions:**

The paper studied a domain generalization setup, where a training domain contains a collection of paired examples that share the same semantic information. They combines two widely-used approaches: data augmentation and representation matching. To achieve the DG goal, they propose a new consistency regularization LAM, which is a variant of logit matching by aligning it for per feature unit individually. The authors empirically compare different regularizations on five datasets and demonstrate the effectiveness of the LAM. They also provide a theoretical framework for their setup.

**Q2-3 Extent To Which Claims Are Supported By Evidence:**

3: Good: the main claims are supported by convincing evidence (in the form of adequate experimental evaluation, proofs, (pseudo-)code, references, assumptions).

**Q2-4 Reproducibility:**

2: Fair: key resources (e.g. proofs, code, data) are unavailable but key details (e.g. proof sketches, experimental setup) are sufficiently well-described for an expert to confidently reproduce the main results.

**Q3 Main Strengths:**

1. Figure 1 is illustrative and informative.
2. The paper is well-organized and clearly written
2. Empirical evaluations are thorough and authors use GradCAM saliency maps to visualize that LAM can better capture shared representations.

**Q4 Main Weakness:**

1. The theoretical framework is loosely related to the method. Besides, the third assumption that the support set of the testing input distribution lies within the support set of training input is too strong.

2. The two setups CR with and without labeled SS PAIRS is confusing. Data augmentation methods used in the paper are designed to keep the semantic meaning of the original image. How can they fit into unlabeled category? Please clarify.

3. A more concerning question is on the setup. The general DG setup including single domain and multi-domains are well-established. Why do we need the special case pointed out by authors? Are there some practical applications?

4. Comparing with logit matching, LAM improves from norm of logits difference to the sum of norm on logits calculated from per representation unit. A careful comparison is needed.  For example, empirically, I can add a higher penalty coefficient for logit matching. Or are there theoretical benefits from the design?

5. A minor issue is on the usage of term 'causal'. Causality in statistical learning has a strict definition and is related to casual structure learning, causal graph discovery, etc. In DG setup, it is ambitious to say that domain-invariant representation is a causal factor of the target.

**Q5 Detailed Comments To The Authors:**

See my above comments on strength and weakness.

**Q9 Complying With Reviewing Instructions:**

Yes

---

> ### Author Rebuttal · Authors · 2024-04-05
>
> We thank the reviewer for their valuable time and constructive feedback. We address the reviewer’s concerns below. Please let us know if there are additional questions and concerns or if all concerns have been addressed.
> ## Clarification for (un)labelled SS pairs
> The distinction between labelled and unlabelled SS pairs is **whether the class labels are directly utilized by the regularizer**. For example, logit matching aligns the logits of *all* classes regardless of the class label of the pair, and thus uses “unlabelled” SS pairs. In comparison, target logit matching and LAM align only the logit w.r.t. the class label of the SS pair which therefore is “labelled”.
>
> This distinction matters because, in practice, ensuring $x$ and $\tilde{x}$ hold the same semantic information for all classes is harder than for a single class. Regularizing in the “labelled” way helps avoid imposing over-stringent constraints on all classes while preserving enough power to enforce causal-invariant prediction. The advantage of using labelled pairs is shown by the iWildCam-N experiments (results in Table 2) where only methods using labelled pairs outperform ERM+DA.
>
> Thanks for the great feedback. We will revise the paper to make the concept of (un)labelled SS pairs clearer.
> ## About our theory
> Yes, our theory is more closely related to consistency regularization (CR) in general. It articulates the sufficient conditions for optimal DG, whereby the second condition connects CR with DG. It is an integral part of the paper to help readers better understand our position -- why we consider CR as a good alternative to other established DG approaches.
>
> We also understand your concern about the third assumption, but the assumption is actually *not* strong relative to the result, namely *optimal* DG. In fact, it is necessary for optimal DG in general problems. In general problems, we can’t assume any relation between $p(y|x^c)$ and $p(y|\tilde{x}^c)$ for $x^c$ and $\tilde{x}^c$ that are different. This is because the function $y = h(x^c)$ could take any form. Hence, if we do not observe $(x, y)$ generated from certain $x^c$, we may not know enough about $p(y|x^c)$, and therefore optimal DG can’t be guaranteed.  The assumption can be relaxed if we consider approximately optimal DG or make some assumptions about $h$. We opt for simplicity since the choice of assumptions about $h$ is not pertinent to the focus of this paper.
> ## Practicality of our setup
> Data augmentation, which naturally gives rise to SS pairs, is commonly used in DG. The pairing information of SS pairs, however, is not fully utilized in the single or multi-source setup, leading to suboptimal results. Under our setup, we explore more effective ways to utilize this information. This proves to be quite beneficial, as shown in Table 3, where our method outperforms the representative methods under other setups.
>
> Also, some tasks come with naturally paired data. For instance, in object recognition, this includes images of the same objects in different environments; or in face recognition, this includes photos of the same person.
>
> ## Difference between TLM and LAM
> We believe you asked about the difference between target logit matching $r_{TLM}$ and logit attribution matching $r_{LAM}$, where:
> $r_{TLM}(x,\tilde{x};y)=[z_θ^{y}(x)-z_θ^{y}(\tilde{x})]^2,$ and
>
> $r_{LAM}(x,\tilde{x};y)=\sum_u[f_ϕ^u(x){{w_u}_y}-f_ϕ^u(\tilde{x}){{w_u}_y}]^2.$
> While TLM directly aligns two logit values, LAM aligns the logit contributions from each feature unit $u$ to the class $y$.
>
> In fact, $r_{TLM}$ is bounded by $r_{LAM}$, following Jensen’s inequality: $$\sum_u[f_ϕ^u(x) {{w_u}_y}-f_ϕ^u(\tilde{x}) {{w_u}_y}]^2 \geq \frac1d\Big[\sum_u f^u_ϕ(x){{w_u}_y}-\sum_u f^u_ϕ(\tilde{x}){{w_u}_y}\Big]^2=\frac1d[z_θ^{y}(x)-z_θ^{y}(\tilde{x})]^2$$ for any $(x,\tilde{x};y)$, where $d$ is the number of feature units.
>
> This inequality tells us that **a smaller $r_{LAM}$ implies a smaller $r_{TLM}$, but not vice versa**. In this sense, LAM is finer-grained than TLM. To answer your question, a higher penalty coefficient for TLM makes no real difference as it is *fundamentally weaker* than LAM.
>
> In our experiments, the same effort has been paid to find the best “penalty coefficient” (λ) for both methods. We tried a wide range of λ values and selected the one with the best validation performance. We provide below the results on iWildCam under different λ for TLM. Optimal TLM OOD is achieved at λ=0.5 with an F1 score of 38.5, yet it falls short of LAM's performance.
> ||||
> |-|-|-|
> ||λ|OOD F1|
> |ERM+DA|N/A|36.5|
> |**LAM**|5|**41.2**|
> |TLM|10|36.2|
> |TLM|5|37.5|
> |TLM|1|38.2|
> |**TLM**|0.5|**38.5**|
> |TLM|0.4|38.3|
> |TLM|0.3|38.1|
> |TLM|0.2|38.3|
> |TLM|0.1|37.9|
> ## The usage of the term “causal”
> Thanks for the comment. We will carefully revise related parts of the manuscript to make the usage of the term “causal” more appropriate.

---

### Official Review · Reviewer_CVC9 · 2024-03-23

**Q2-1 Originality-Novelty:** 2
**Q2-2 Correctness-Technical Quality:** 2
**Q2-5 Clarity Of Writing:** 3

**Q10 Ethical Concerns:**

none noted.

**Q1 Summary And Contributions:**

the paper introduces a new method for domain generalization that leverage consistency regularization over the samples and its augmented pairs. The paper has some interesting theoretical discussions, but the empirical scope is too limited, and the novelty is kind of limited.

**Q2-3 Extent To Which Claims Are Supported By Evidence:**

3: Good: the main claims are supported by convincing evidence (in the form of adequate experimental evaluation, proofs, (pseudo-)code, references, assumptions).

**Q2-4 Reproducibility:**

4: Excellent: key resources (e.g. proofs, code, data) are available and key details (e.g. proof sketches, experimental setup) are comprehensively described for competent researchers to confidently and easily reproduce the main results.

**Q3 Main Strengths:**

1. the paper offers an interesting discussion of domain generalization where the data has paired samples.

2. it also introduces an extensive discussion of theoretical results.

**Q4 Main Weakness:**

1. the main problem is the novelty of this work. Similar work has been done in a more comprehensive manner [1]. The consistency loss has been discussed extensively, see the related work of [1] in various settings, and [1] compared multiple consistency loss and its performances in several robustness settings.

2. the theoretical discussion is also fairly trivial, for example, the model is "causal invariant" is already part of the hypothesis

3. the emprical scope is also limited, there are multiple more power DG methods that has not been discussed. There are not methods compared in the most recent two years.

[1]. Wang, Haohan, et al. "Toward learning robust and invariant representations with alignment regularization and data augmentation." Proceedings of the 28th ACM SIGKDD Conference on Knowledge Discovery and Data Mining. 2022.

**Q5 Detailed Comments To The Authors:**

please see the weakness above.

**Q9 Complying With Reviewing Instructions:**

Yes

---

> ### Author Rebuttal · Authors · 2024-04-05
>
> We thank the reviewer for their valuable time and constructive feedback. We address the reviewer’s concerns below. Please let us know if there are additional questions and concerns or if all concerns have been addressed.
> ## About novelty
> We thank the reviewer for pointing out a related work [1] to us. We will acknowledge and discuss the work in our paper in a faithful and accurate account. **We must, however, respectfully *disagree* that our work has novelty issues when compared to [1].** Below we highlight their differences and explain why our contribution is novel and significant.
>
> **First, the main contribution of the two papers is totally different.** Wang et al. [1] evaluated various pre-existing ways to implement logit matching and concluded logit matching is most effective with l2-norm (among some other distance measures). While their evaluation is extensive and meticulous, they did *not* introduce new consistency regularization (CR) method. In contrast, we propose a novel CR method called LAM and show that it outperforms logit matching with l2-norm and other previous CR methods (e.g., feature/probability matching) not covered in [1].
>
> **There are also significant differences between the experiment settings of the two works:**
>
> 1. CR methods evaluated: [1] evaluated different metrics for quantifying the disparity between two **logit vectors**: l1-norm, l2-norm, cosine similarity, etc. In our experiments, we considered a much larger set of regularization targets including **features**, **logits**, and **output probabilities**. More specifically, we evaluated logit/feature matching with l2-norm, probability matching with KL/JS divergence, target probability/logit matching with l2-norm, and our logit attribution matching (LAM). There is **little overlap** between the set of CR methods evaluated in the two papers.
>
> 2. Domain shifts considered: [1] focused on **predefined synthetic** domain shifts, e.g., synthetic texture bias, image rotation, and contrast adjustment. In comparison, we focus on **complex, natural** domain shifts in **real-world** scenarios, as those in NICO, iWildCam, and Camelyon.
>
> 3. Data augmentation (DA) methods used: [1] applied DA that is directly informed by the controlled synthetic shifts. In contrast, we explore both task-agnostic, generic DA - RandAugment [10]; as well as task-specific, targeted DA [11] to more comprehensively evaluate the CR methods under realistic conditions.
> ## Theoretical discussion
> We would like to clarify that the theoretical discussion is to motivate CR as a general approach to DG, not to make a significant theoretical contribution. The theory articulates the sufficient conditions for optimal DG, whereby the second condition (the causal-invariant condition) connects CR with DG. It is an integral part of the paper to help readers better understand our position -- why we consider CR as a good alternative to other established DG approaches.
> ## Empirical scope
> Most DG methods we considered are widely regarded as strong DG baselines. These methods are adopted by incumbent DG benchmarks well recognized by the community [2, 3]. From the benchmarks, it is also quite noticeable that ERM remains a very competitive baseline. Very few DG methods, including the recent ones, stand out.
>
> Current SOTA-level DG methods are mostly model ensemble methods [4,5,6] and fine-tuning techniques for pre-trained models [7,8,9]. These methods are **orthogonal** to LAM and hence were not discussed in the paper.  In fact, LAM can seamlessly combine with those methods to further improve OOD performance.
>
> We have combined LAM with a SOTA-level fine-tuning method, FLYP [7], and compared the results with SOTA model ensemble methods.  FLYP is a method to improve the fine-tuning process of vision-language models like CLIP for classification tasks. We conducted experiments on the iWildCam dataset with the results shown below. The performance of FLYP+DA was obtained by incorporating target-augmented images in the FLYP training process, while FLYP+DA+LAM is our method which applies the LAM regularization to the original and augmented image pairs.
> | Method|Backbone|ID F1$\uparrow$|OOD F1$\uparrow$|
> |-----------------|--------------------|-------|---------|
> |FLYP+DA+LAM|CLIP ViT-L/14@336|60.9|**48.7**|
> |FLYP+DA|CLIP ViT-L/14@336|58.9|47.1|
> |FLYP [7]|CLIP ViT-L/14@336|59.9|46.0|
> |WiSE-FT [6]|CLIP ViT-L/14@336  | 55.8|46.4|
> |ERM|CLIP ViT-L/14@336|52.1|39.9|
> |||||
> |FLYP+DA+LAM|CLIP ViT-L/14|59.0| **45.6**|
> |FLYP+DA|CLIP ViT-L/14|59.0|44.3|
> |FLYP  [7]|CLIP ViT-L/14| 56.9|43.4|
> |Model Soup [5]|CLIP ViT-L/14|57.6|43.3|
> |ERM|CLIP ViT-L/14|55.8|41.4|
>
> The results show that LAM significantly improves the model's OOD performance over both FLYP and FLYP+DA. In addition, LAM (without model ensemble) outperforms the ensemble methods, Model Soup [5] and WiSE-FT [6]. We will include the results in the final version of the paper.
>
> [References](https://tinyurl.com/46exk5mj)

---

### Meta-Review · Area_Chair_p45o · 2024-04-16

The four reviewers did not agree, but there was a noticeable tendency in favour, with three reviewers in favour (bXye, p8b3, 97yb) and one strongly against (CVC9). The post-rebuttal discussion helped to clarify the concerns and overall the strengths and weaknesses of the work. One of the reviewers in favour, indeed the one who initially gave this work the highest score (p8b3), decided to lower their score after the discussion, to acknowledge the concerns that had been flagged by other reviewers. On the other hand, the opposing reviewer (CVC9) asserted their opinion and remaining concerns about this work.

Upon a quick look at the paper, I agree with the reviewers that this work has strengths and contributions to merit publication at UAI. I confirm that the paper is mostly "well-organized and clearly written" and "easy to follow and makes its contributions clear" as expressed by two reviewers. Likewise, I confirm that "the authors investigate several possible variants of the regularisation penalties" and they also "acknowledge limitations of their work regarding the scope and strength of assumptions" to some extent. There are other expressions in favour regarding the theoretical section and the empirical evaluations. At the same time, as said above, the reviewers have pointed out some concerns about this work, which have made it clear that significant improvements would need to be made if this work were to be accepted at this time. The concerns about novelty suggest that the authors need to make changes such as to make it clear to readers how their work is positioned with respect to the related literature. Besides discussions of the related literature including Wang et al. 2022, a related concern that the authors should compare to more baselines including recent works such as these:

- Berezovskiy, Valeriy, and Nikita Morozov. "Weight Averaging Improves Knowledge Distillation under Domain Shift." arXiv preprint arXiv:2309.11446 (2023).

- Huang, Zeyi, et al. "The two dimensions of worst-case training and their integrated effect for out-of-domain generalization." Proceedings of the IEEE/CVF Conference on Computer Vision and Pattern Recognition. 2022.

I would like to highlight also the need to make significant improvements in the theory section, to address the comments of the reviewers. I agree with the concerns regarding the use of the term "causal" which could lead to confusions in current form. The language (terminology) and notation needs to be cleaned-up accordingly. Personally, I found the definition "causal-invariant" to be a potential source of confusion, since the invariance in this definition is with respect to the non-causal factors. The authors should consider choosing a different term for this concept of invariance. The upper script $d$ in the notation $P^d$ is unnecessary, I think. Perhaps the upper scripts "$c$" and "$n$" should be in text (roman) font, to see $X^{\text{c}}$ and $X^{\text{n}}$ and similar for other, since these letters aren't variables. There is a strange mix up of the use of capitals and lower cases for random variables and their possible values: I support the use of say $X$ to denote the random variable, $x$ an arbitrary possible value of this variable, and $\mathcal{X}$ the set of all possible values of this variable; however when declaring measures such as probabilities in terms of densities it is more natural to write $P(x)$ than $P(X)$, is it not? Similar for the conditional probabilities. In any case, choose one convention and stick to it throughout the paper. A quick look at equations (1) and (2) makes it evident that the said mix up of capitals and lower case, i.e. mixing up the conventions. I strongly suggest to avoid using footnote markers next to mathematical notation! In particular, for footnotes 2 and 3, find a different way to place the footnote marker. I would like to add these requests: Check the formatting of the references, to ensure correct capitalisation in the titles, e.g. to read iWildCam, RandAugment, ImageNet, AugMix, WILDS, REx, GrabCut, Grad-CAM, FixMatch, H&E, PHH3, CutMix, etc. Also check to ensure capitalised and consistent venue names (conferences).

If this paper were to be accepted, I strongly request that authors respect the promises they made in the rebuttal, namely the additional discussions and clarifications, as well as fixing the issues that were pointed out in the reviews. I would encourage the authors in this case to do this conscientiously. Last but not least, I would encourage the authors in this case to additinally carry out a through proof reading to ensure that the final version of the paper is of high quality.